# The nucleoid of rapidly growing Escherichia coli localizes close to the inner membrane and is organized by transcription, translation, and cell geometry

Christoph Spahn [1,2,3] ✉, Stuart Middlemiss [4], Estibaliz Gómez-de-Mariscal[5,6,7], Ricardo Henriques [5,7,8], Helge B. Bode[2,9,10,11,12], Séamus Holden [4,13] & Mike Heilemann [1] ✉

Bacterial chromosomes are spatiotemporally organized and sensitive to environmental changes. However, the mechanisms underlying chromosome configuration and reorganization are not fully understood. Here, we use single-molecule localization microscopy and live-cell imaging to show that the Escherichia coli nucleoid adopts a condensed, membrane-proximal configuration during rapid growth. Drug treatment induces a rapid collapse of the nucleoid from an apparently membrane-bound state within 10 min of halting transcription and translation. This hints toward an active role of transertion (coupled transcription, translation, and membrane insertion) in nucleoid organization, while cell wall synthesis inhibitors only affect nucleoid organization during morphological changes. Further, we provide evidence that the nucleoid spatially correlates with elongasomes in unperturbed cells, suggesting that large membrane-bound complexes might be hotspots for transertion. The observed correlation diminishes in cells with changed cell geometry or upon inhibition of protein biosynthesis. Replication inhibition experiments, as well as multi-drug treatments highlight the role of entropic effects and transcription in nucleoid condensation and positioning. Thus, our results indicate that transcription and translation, possibly in the context of transertion, act as a principal organizer of the bacterial nucleoid, and show that an altered metabolic state and antibiotic treatment lead to major changes in the spatial organization of the nucleoid.

Chromosome replication and segregation represent the fundamental processes underlying cell proliferation and survival. In bacteria, they occur simultaneously with other cellular processes, such as transcription, translation or signaling, and share the same space due to the lack of compartmentalization. Despite this lack of compartmentalization, bacterial chromosomes were found to be highly organized spatiotemporally, with nucleoid organization strongly differing between bacterial species[1–3]. The entity of chromosomal DNA in a bacterial cell is called the 'nucleoid'' and much research is dedicated to investigate its organization. Seminal work showed that the *Escherichia coli* nucleoid shows structuring into domains of varying sizes on the molecular level by the action of nucleoid-associated proteins[4–6].

Similar to eukaryotic nuclei, chromosomal regions populate specific areas and show precise intracellular positioning[7,8]. Interestingly, the *E. coli* nucleoid exhibits varying complexity, i.e., degree of sub-structuring, depending on the nutrient availability and, thus, growth rate[9]. In particular, nucleoids are structurally more complex during fast growth than during slow growth, where it populates a larger fraction of the cytosol. The global morphology of these 'complex' nucleoids was found to be stable on the minute time scale, while fast imaging in slowly growing cells revealed rapid transversal nucleoid fluctuations along the entire bacterial long axis[8,9,10]. Despite these large differences in organization, we still do not fully understand the molecular mechanisms that shape the nucleoid in an environment-dependent manner.

One major driver of chromosome organization is entropy, which contributes to sister chromosome segregation, nucleoid positioning, and compaction[11–13]. Other factors involved in nucleoid organization are confinement, molecular crowding, cell size and morphology[13,14]. Wall-less *B. subtilis* cells, so-called L-forms, exhibit abnormal cell and nucleoid shapes, while their growth in a confined space resulted in nucleoid morphology and chromosome segregation patterns as they are seen in walled cells[15]. Chromosome size and positioning was further found to scale with cell size due to confinement and molecular crowding[13]. The correlation between cell and nucleoid size was recently extended to a wide range of bacterial species, highlighting a conserved contribution of physical principles to chromosome organization[16]. Solvent quality of the bacterial cytosol is another physical factor that can explain the distribution of the nucleoid in *E. coli*[17].

In addition to physical effects, various biological processes strongly influence bacterial chromosome organization. A large body of work investigated the effect of biosynthetic processes such as transcription or translation on nucleoid structure, revealing dramatic reorganization during inhibition[18–24]. These effects are so severe that cell and nucleoid morphology can be used as a readout for drug mode-of-action studies, drug screening applications and antibiotic susceptibility testing[25–27]. Particularly, transcription was found to affect chromosome size and dynamics in different bacterial species as shown by Hi-C and microscopic experiments[17,18,28–30]. Inhibition of translation or DNA replication also leads to strong nucleoid phenotypes, indicating that chromosome organization is tightly connected to biosynthetic processes[18–20,22]. Of note, nucleoid reorganization by transcription- and translation-halting drugs occurs on a rapid time scale, as e.g., shown by Bakshi and colleagues[19]. This provided evidence that spatiotemporal coupling of transcription, translation, and insertion of membrane proteins, so-called transertion, keeps the nucleoid in an expanded state[19,21]. Localized protein biosynthesis increases efficiency and is conserved in eukaryotic cells, in which proteins are co-translationally translocated into the endoplasmic reticulum[31]. As transcription in bacteria often occurs co-translationally[32], an indirect coupling of the nucleoid to the membrane represents an attractive hypothesis[33–35]. In fact, passive segregation by membrane attachment of the origin of replication together with cell elongation was one of the first proposed mechanisms for bacterial chromosome segregation[36]. The observation of rapid re-centering of nucleoids in asymmetrically dividing cells, however, highlights entropy as the driving force behind chromosome positioning[15]. While transertion could just be a consequence of co-transcriptional translation in a highly confined space, it was recently shown to be important for the assembly of bacterial secretion systems[37].

Most microscopic work on bacterial chromosome organization used diffraction-limited approaches and do not take advantage of the benefit provided by super-resolution microscopy. Several studies employed 3D structured illumination microscopy (SIM) to investigate nucleoid structure in live *E. coli*[38,39] and *B. subtilis* cells[40], highlighting differences in nucleoid morphology between species and growth conditions. In particular during fast growth, the *E. coli* nucleoid was found to have a more complex morphology than the *B. subtilis* nucleoid. Single-molecule localization microscopy (SMLM)[41,42], however, is rarely used to study bacterial chromosome biology, although its superior resolution provided valuable insights into other processes, such as cell-wall synthesis or cell division[43,44]. We thus sought to study *E. coli* nucleoid organization with SMLM, both during unperturbed growth and drug treatment. We chose rapidly growing cells to investigate the highly complex and structured nucleoid[9] and to reveal the mechanisms that mediate this organization. SMLM showed a condensed nucleoid positioned close to the inner membrane, which we could verify in live cells. To investigate the nature of this organization, we chemically fixed bacteria at different time points during inhibition of cell wall synthesis, transcription, translation, protein translocation and DNA replication. Perturbation of cell wall synthesis highlighted the membrane-proximal state of the nucleoid. Inhibition of transcription or translation, on the other side, resulted in a rapid nucleoid repositioning away from the membrane within the time scale of 2–10 min, suggesting transertion as the mechanism behind the observed nucleoid anchoring. By reversibly arresting DNA replication, we further observe that nucleoids locate at the cell center of elongating cells and rapidly repopulate DNA-free areas after release from replication block. This happens faster than cell elongation, highlighting the role of cell geometry and entropy in nucleoid positioning and expansion. Together, our work pinpoints to transertion as a determinant of the three-dimensional configuration of the *E. coli* nucleoid during fast growth, while nucleoid positioning is mainly mediated by entropic effects.

## Results

### The nucleoid in fast-growing *E. coli* is condensed and positioned close to the membrane

Previous studies suggested that the nucleoid in fast-growing *E. coli* possesses a helical or twisted configuration[6,9,10]. To investigate this at high spatial resolution, we metabolically labeled DNA in *E. coli* cells growing at a mass doubling time of 27 min using 5-ethynyl-2-deoxyuridine (EdU)[45]. We labeled these cells with Alexa Fluor 647 via click chemistry, which allows for high-resolution 3D single-molecule imaging (Fig. 1A) due to the superior brightness of this fluorophore. 3D imaging revealed a condensed, ring-like nucleoid structure with a clear DNA-free center in the cross-section (Fig. 1Aiii, iv, Supplementary Movie 1). This arrangement was not seen in the 2D projection (Fig. 1Ai), highlighting that projection effects in widefield and 2D SMLM images might not represent the actual organization of the target structure[7,46]. In the 3D image, the nucleoid appears positioned close to the cell membrane, as we have previously shown using STED and 3D SIM microscopy[47,48]. To test whether these contact sites correlate with membrane protein assemblies, we co-imaged the nucleoid with MreB, which is part of the elongasome, an essential, abundant and exclusively membrane-associated multi-protein complex. As copper-catalyzed click-labeling destroys fluorescent proteins, we used transiently binding labels and performed dual-color point accumulation for imaging in nanoscale topography (PAINT) in cells expressing an MreB^sw-superfolderGFP (MreB^sw-sfGFP) protein fusion[23,49]. To increase the resolution, we deconvolved 3D stacks of the MreB fusion (Fig. S1) and registered the resulting image with the highly-resolved PAINT images. These images suggest a moderate degree of spatial correlation between elongasomes and the nucleoid, which is investigated in this study.

To exclude fixation artefacts and validate that the membrane-proximal nucleoid arrangement exists in vivo, we created *E. coli* strains that express an H-NS-mScarlet-I or HU-α-mScarlet-I protein fusion in the MreB^sw-sfGFP background (Tables S1 and S2). To obtain images of cross-sections, we imaged these strains in vertical orientation using the VerCINI approach (Fig. 1C)[50,51]. While the nucleoid is positioned at the radial cell center during slow growth (M9, $t_d$ ~ 120 min), it locates at the periphery during rapid growth, both at 30 °C and 37 °C

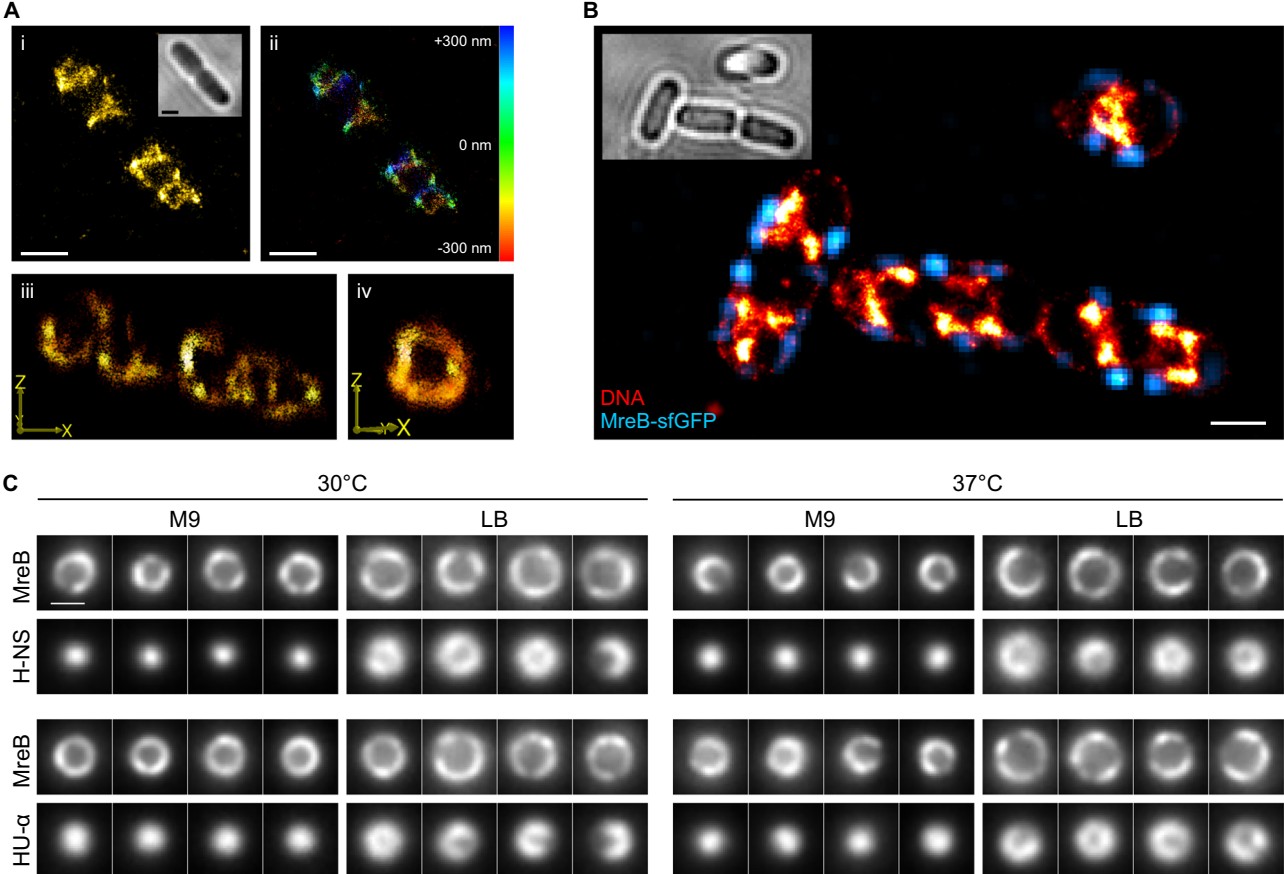

**Fig. 1 | The nucleoid of fast-growing *E. coli* positions along the inner membrane.** **A** 3D dSTORM images of a metabolically labeled (EdU, clicked with Alexa Fluor 647), chemically fixed cell shown as 2D projection (i), color-coded by depth (ii) and tilted 3D views (iii, iv). **B** 2D PAINT image of the nucleoid and membrane (red hot) of a fixed NO34 cell expressing an MreB[sw]-sfGFP fusion (deconvolved, cyan). **C** Live-cell VerCINI[50,51] measurements of *E. coli* cells expressing MreB[sw]-sfGFP and H-NS/HU-α-mScarlet-I fusions. Shown are cross-sections of individual cells. Nucleoids are condensed in the radial cell center when cells are grown in M9 ($t_d$ ~ 100 min), but close to the membrane during fast growth in LB ($t_d$ ~ 27 min). Scale bars are 1 μm (**A**, **B**) and 500 nm (**C**).

($t_d$ ~ 25–30 min). VerCINI imaging of DAPI-stained cells verified a low DNA-density at the cell center (Fig. S2). We thus deduced that the highly twisted arrangement and membrane-proximal positioning is a feature of the *E. coli* nucleoid during rapid growth. The increased nucleoid complexity at faster growth is in agreement with previous studies, so is the centered positioning of an elongated nucleoid during slow growth[9,10]. We also performed live-cell imaging of these strains using confocal laser-scanning microscopy (CLSM), which showed nucleoid dynamics on the second time scale and a stable nucleoid positioning at the cells' quarter positions (Supplementary Movies 2 and 3). Global nucleoid positioning hereby coincided with MreB distribution (Fig. S3).

## Inhibition of biosynthetic processes alters nucleoid organization

To study the basis for the observed membrane-proximal positioning, we treated exponentially growing *E. coli* cultures with antibiotics that inhibit specific steps in biosynthetic pathways (Fig. 2). For inhibition of cell wall synthesis, we used MP265 (A22 analog, inhibits MreB polymerization)[52] and mecillinam (PBP2 inhibitor, inhibits peptidoglycan transpeptidation during cell elongation)[53]. Protein biosynthesis was inhibited using high concentrations of rifampicin (transcription inhibition) and chloramphenicol (translation inhibition)[19,54]. Additionally, we perturbed protein translocation (sodium azide)[55] and DNA supercoiling, the latter leading to inhibition of DNA replication (nalidixate, DNA gyrase and topoisomerase IV inhibitor)[56]. Each antibiotic was added to the *E. coli* culture during mid-

log phase ($OD_{600}$ ~ 0.25) and aliquots were chemically fixed at defined time points, labeled for DNA and membranes and imaged by CLSM (Fig. 2A). Over the time course of 60 min during unperturbed growth, the nucleoid shows the expected sub-structuring and positioning at quarter positions (Fig. 2B)[9,45]. Both MP265 and mecillinam treatments induced a rod-to-sphere transition with significant cell rounding occurring after 30 min. Interestingly, inhibition of MreB polymerization with MP265 led to the rapid formation of polar foci or aggregates that were most prominent in the first 2–10 min of treatment and successively disappeared with ongoing MP265 exposure. Azide treatment did not show apparent changes in nucleoid morphology, but increasing MreB aggregation at 30–60 min exposure. When we inhibited protein biosynthesis, changes in nucleoid morphology were more dramatic. Both inhibition of transcription (rifampicin) and translation (chloramphenicol) led to an instant contraction of the nucleoid in the first minutes, in line with previous work[19]. While nucleoids started to expand after ~10–20 min of rifampicin treatment, condensation continued in chloramphenicol-treated cells. Stalling of replisomes by the gyrase inhibitor nalidixate led to nucleoid positioning at the cell center, while cells continued to grow and formed large, nucleoid-free areas. MreB hereby populated the entire cell cylinder and facilitated elongation also in nucleoid-free regions.

To assess global changes induced by the different antibiotics, we developed an image analysis routine that provides heat maps of the nucleoid and MreB (Fig. 3). For this purpose, we extracted individual cells from CLSM images and processed these images by straightening, alignment, normalization and finally averaging (Fig. 3A, i). The

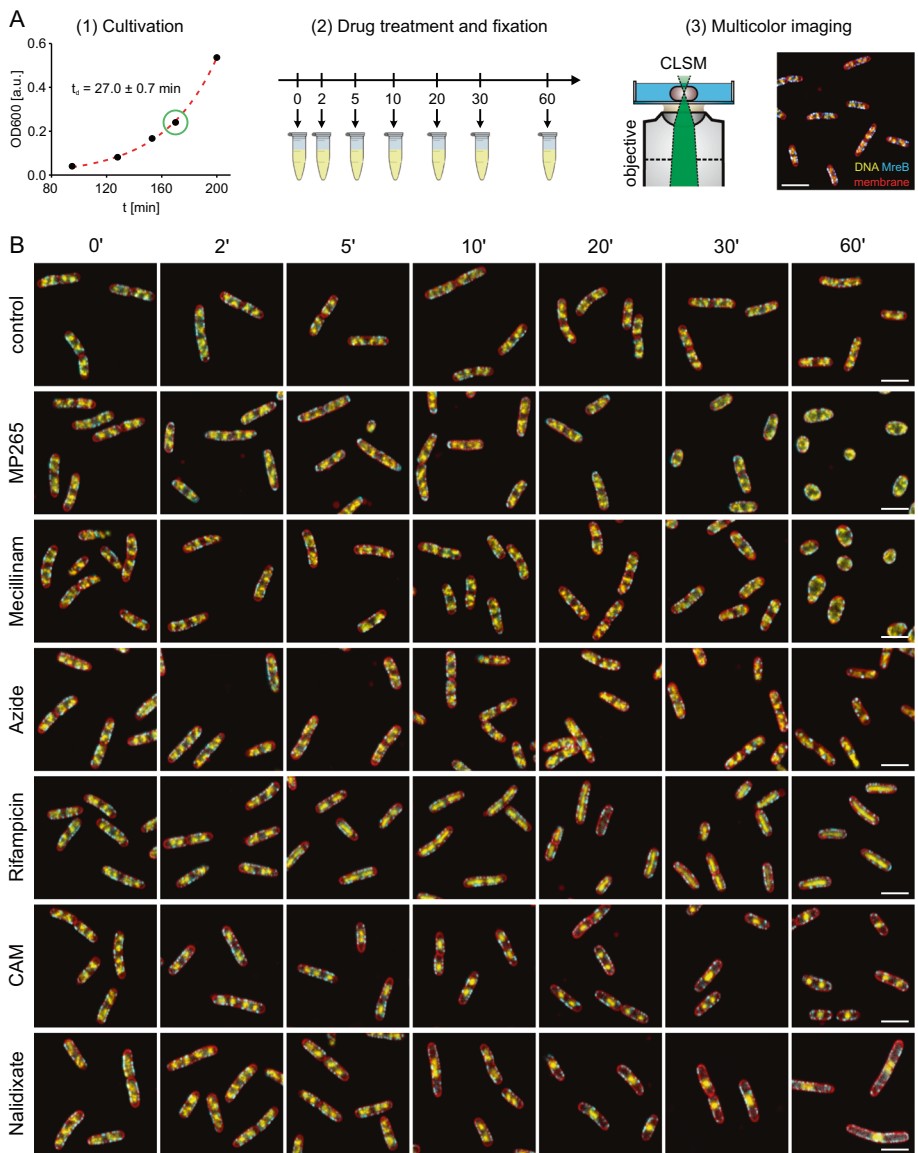

**Fig. 2 | Cell fixation allows capturing antibiotic action dynamics at the minute time scale. A** Schematic of the experimental design. Cells were grown in LB to exponential phase ($t_d$ ~ 27 min), antibiotics were added and aliquots were chemically fixed at distinct time intervals. Bacteria were then immobilized, stained for DNA (DAPI, yellow hot) and membranes (Nile Red, red) and imaged using CLSM. Chromosomally expressed MreB-sfGFP$^{sw}$ is shown in cyan. **B** Representative images of bacteria treated with different antibiotics for specific time intervals. CAM = Chloramphenicol. Scale bars are 3 μm.

resulting images represent the average cell for the specific condition (antibiotic treatment and exposure time), which we term 'population average' throughout this study. Information on the number of cells used for averaging as well as information on cell length is provided in Table S3.

For quantitative analysis, we measure the relative nucleoid length (RNL) in individual cells (Fig. 3A, ii). RNL describes the fraction of cell length that is populated by chromosomal DNA. This metric is similar to the relative nucleoid size used by Cabrera and colleagues[54], yet we prefer RNL in diffraction-limited images due to the 2D projection effects shown in Fig. 1A and literature[7,46]. To monitor reorganization of MreB following antibiotic treatment, we quantified its relative distribution at the cell cylinder and cell poles (Fig. 3A, iii). The latter are typically avoided by MreB, likely due to a combination of processive elongasome-driven motion of MreB and spontaneous alignment of curved MreB filaments circumferentially around the cell sidewall[57]. Population averages of the drug treatments validate the observations made in Fig. 2. Untreated cultures showed two bilobed sister chromosomes, as it is expected under fast-growing conditions (Fig. 3B)[9].

While population averages hide length-dependent effects, the existence of two nucleoids and the global positioning at quarter positions was observed for all length intervals (Fig. S4). Upon inhibition of MP265 and mecillinam treatment, changes in nucleoid positioning and morphology only appeared during rod-to-sphere transition, while apparently, no changes were observed during short drug exposure. To assess global changes in nucleoid organization, we measured the long- and cross-axis profiles in population averages (Figs. S5–S8). Polar recruitment of MreB during MP265 treatment is clearly visible in length-axis plots (Fig. S5) and was strikingly consistent for both replicates. Concomitant with cell rounding, the bilobed nucleoid distribution diminished for both MP265 and Mecillinam treatment (Fig. S5). Here, cross-axis plots did not reveal strong reorganization along the bacterial short axis (Fig. S6).

For the inhibition of protein biosynthesis, population averages revealed severe changes in nucleoid morphology. When inhibiting transcription (rifampicin) or translation initiation (chloramphenicol), population averages show an abrupt nucleoid condensation along the bacterial long axis. Nucleoids expand again after 5 min rifampicin

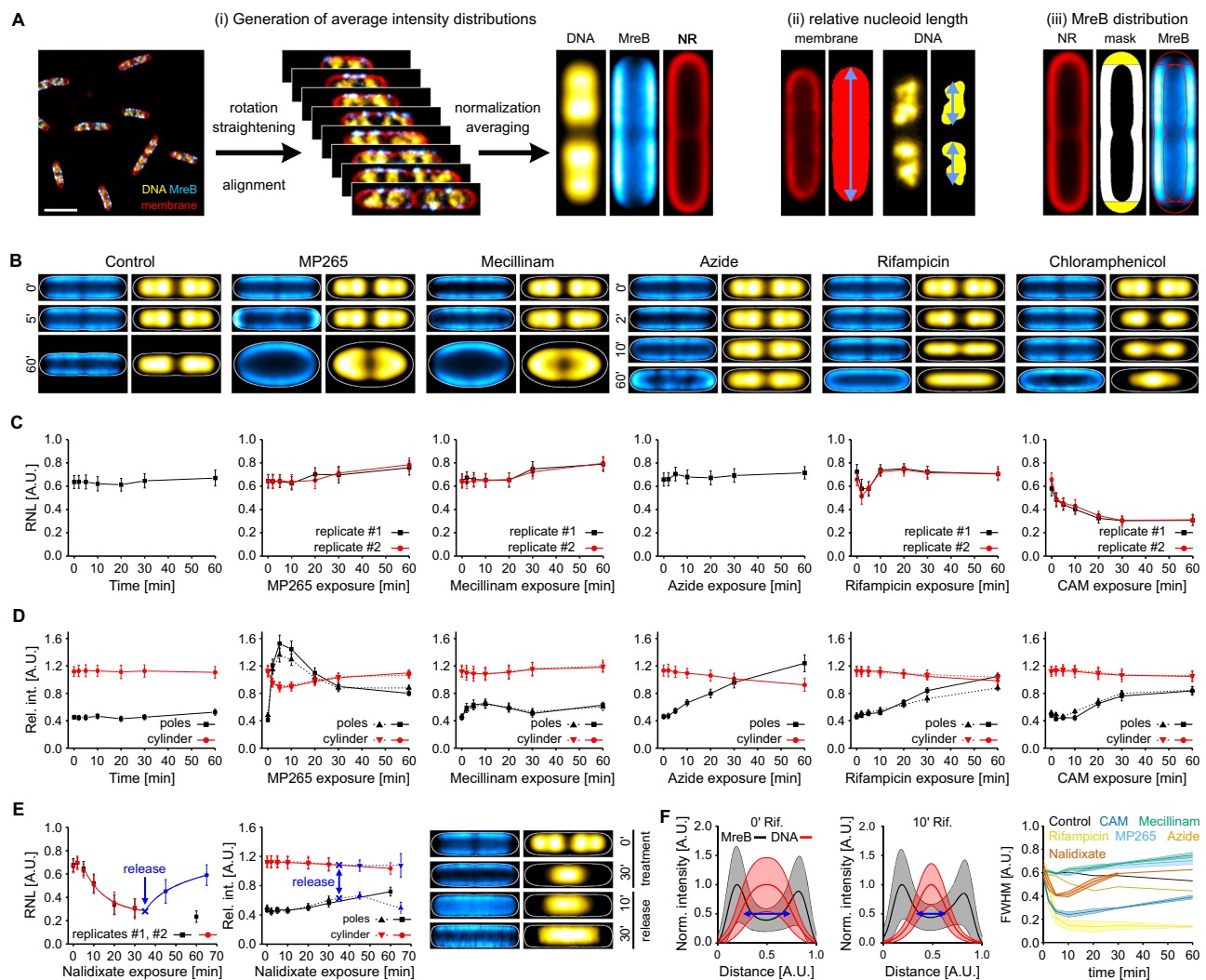

**Fig. 3 | Analysis of drug-treated *E. coli* cells on the population- and single-cell level. A** Analyses performed with CLSM images. DNA is displayed in yellow, MreB^(SW)-sfGFP in cyan and the membrane in red. Schematic of the generation of population average images. (i) Bacterial cells are detected in confocal images, rotated, straightened and aligned. After normalization by cell length and width, cells are averaged for each time point, generating a population average image (see methods). NR = Nile red. (ii) DNA distribution along the bacterial long axis was determined on the single cell-level by measuring the nucleoid length relative to the cell length. (iii) Quantification of MreB distribution: Polar (yellow) and cylindrical sections (white) were segmented in population average images and the relative MreB intensity was calculated (see methods). **B** Population average images of selected time points (control: *N* = 121–213; MP265: *N* = 125–231; Mecillinam: *N* = 90–270; Azide: *N* = 136–225; Rifampicin: *N* = 179–364; CAM: *N* = 128–219; Nalidixate: 126–204 (treatment) and 34–76 (release)) Individual cell counts and cell lengths are provided in Table S3. Cell outlines (white lines) were obtained from membrane average images. **C** Relative nucleoid length (RNL) and **D** MreB intensity distributions of antibiotic treatments shown in (**B**). **E** Relative nucleoid length (left),

MreB intensity distribution (middle) and selected population averages (right) of replication arrest and release using nalidixate. **F** Determination of the average nucleoid width using population average intensity distributions. Shown are exemplary cross-axis plots without treatment (left) and 10 min rifampicin treatment (middle). The full-width at half maximum (FWHM, blue arrows) was determined by integrating the intensity distribution. FWHM over treatment time differ for antibiotic treatments (right). Error bars in (**C, E**) (left plot) represent the standard deviation and in (**D, E**) (right plot) the standard error of the mean. Control and azide treatment experiments were only performed once while all other experiments were conducted as biological duplicates. Rif = rifampicin, CAM = chloramphenicol. Shaded areas in (**F**) represent standard deviations (left and middle plot: pixel-wise standard deviation of population average images; right plot: standard deviation of two replicates). Lines in (**C, D, E**) are added for visualization purpose and do not refer to an underlying model. Error bars in (**C, D, E**) report on the cellular heterogeneity of individual cultures and do not to refer to replicates. Instead, replicates are shown independently. Scale bar in (**A**) is 3 μm. Source data are provided as a Source Data file.

exposure, while they continue condensing during chloramphenicol treatment (Figs. 3B and S7). Cross-axis plots further showed that nucleoids also contract along the short axis. This effect is stronger for rifampicin treatment than for chloramphenicol (Figs. 3B and S8). Inhibition of protein translocation with sodium azide only induced MreB clustering at long exposure times, while global nucleoid organization was not affected (Figs. 3B, S7, and S8).

RNL measurements (Fig. 3C) and analysis of MreB intensity distribution (Fig. 3D) confirmed our observations made by visual inspection. In particular, RNL analysis showed a maximal longitudinal

nucleoid condensation in rifampicin-treated cells after 5 min, while it plateaued at 30 min in CAM-treated cultures. Other treatments did not show severe changes in RNL. Of note, we did not observe changes in control cultures (Fig. 3C, Fig. S9), which showed a constant RNL of 0.64 ± 0.02 (s.d.) throughout the time course of 1 h. The RNL at *t* = 0 min for all treatments was also reproducible with an average value of 0.65 ± 0.04 (s.d., *n* = 12). MreB distribution analysis revealed maximal polar localization after 5 min of MP265 treatment, which declined gradually with continuous exposure time as observed previously[58]. This effect was also observed in live cells immobilized on

MP265-containing agarose pads (Supplementary Movie 4). However, drug treatment dynamics were delayed on agarose pads, likely due to the change in temperature (25 °C vs 32 °C), lack of agitation, or partial drug inactivation during its addition to warm LB-agar. To exclude artefacts induced by cell density or length, we analyzed the MreB distribution in population averages of the control culture and for specific length intervals (Fig. S10). Both controls showed a constant MreB intensity distribution, with most signals being detected in the cylindrical part of the average cell. Interestingly, inhibition of PG crosslinking using mecillinam also induced a slight reorganization of MreB towards the cell poles (Fig. 3D). Perturbation of protein synthesis or translocation led to a gradual shift of MreB towards the poles. As MreB localization coincides with nucleoid distribution along the bacterial long axis, we additionally blocked DNA gyrase using nalidixate, which is known to stall DNA replication (Fig. 3E). Consistent with previous work, this led to a mid-cell-positioned nucleoid while cell growth continued, resulting in a continuous drop in RNL (Fig. 3E, Figs. S11 and S12)[22]. The average nucleoid length hereby decreased within the first 20 min, caused by division of cells with already separated sister chromosomes (Fig. S11).

To track the cellular response upon the continuation of chromosome replication, we removed nalidixate in one replicate after 30 min treatment by washing cells twice with fresh LB medium. This resulted in a fast increase in RNL following a logarithmic function (Fig. 3E, blue points and line, Fig. S12).

Finally, we quantified nucleoid distribution along the bacterial cross-axis by determining the full-width at half maximum (FWHM) in intensity plots of population averages (Fig. 3F). Nucleoids contracted rapidly during rifampicin and chloramphenicol treatment, reaching a maximal condensation after 5–10 min, while other treatments showed only minor changes over time.

## Super-resolution microscopy reveals nucleoid reorganization from the inner membrane towards the cell center upon inhibition of protein biosynthesis

Recent advances in super-resolution imaging allow visualizing the nucleoid in individual cells at ~30 nm resolution[23,59]. We used transiently binding fluorophores Nile Red and JF$_{646}$-Hoechst in PAINT[60] to provide highly resolved snapshots of drug-treated cells (Fig. 4). The obtained snapshots provide nano-scale structural information on the nucleoid architecture which is inaccessible from diffraction-limited microscopy data (Fig. S13). A collection of cells of different lengths for all treatments are provided in Figs. S14–S19. From these super-resolved images, we quantified the subcellular nucleoid distribution by analytically determining the relative DNA content in consecutive radial layers reaching from the cell periphery (membrane) towards the cell center (Fig. 4A). This analysis results in the radial intensity distribution (RID) of the nucleoid, which we calculated for all treatments and time points (see Table S4 for cell counts).

In agreement with our previous work, super-resolved nucleoids of untreated cells were bilobed and sub-structured, avoiding the cell poles and spanning the entire width of the bacterial cell (Fig. 4B). This image represents the 2D projection of the membrane-proximal nucleoid positioning observed in Fig. 1. Multicolor PAINT imaging in cells expressing H-NS-mScarlet-I shows a good agreement between H-NS and PAINT signals, indicating that we image the entire nucleoid (Fig. S20). Image acquisition with an alternative commercial microscope for super-resolution imaging (Zeiss Elyra PS1) (Fig. S21) led to similar results. Together with live-to-fixed controls that we performed in previous work[23], we are confident that we capture snapshots of the native nucleoid organization occurring in live *E. coli* cells.

The rod-to-sphere transition induced by perturbation of cell wall synthesis resulted in widened cells with expanded nucleoids. This reduces the effect of 2D projection compared to untreated cells, as the DNA is distributed within a larger axial range while the observed cross-

section remains constant (see Supplementary Note 1). Positioning of chromosomal DNA close to the membrane thus becomes more evident (Fig. 4B, MP265, and mecillinam, 30' and 60'). In contrast to untreated cells ($t = 0$ min), the nucleoid also populated the cell poles, indicating that cell geometry affects nucleoid positioning. As observed in CLSM images (Figs. 2 and 3), treatment with sodium azide did not lead to global nucleoid reorganization.

Next to MreB aggregation, nucleoids appeared to be slightly less condensed, which might be attributed to a reduced activity of SMC complexes upon de-energization. Specific inhibition of SecA by sodium azide explains MreB reorganization, as a similar MreB mislocalization was recently observed in SecA-defective cells[61]. PAINT images of rifampicin- and chloramphenicol-treated cells visualized the rapid collapse of the nucleoid upon inhibition of protein biosynthesis (Fig. 4B). Here, nucleoids remained partially positioned in proximity of the membrane within the first minutes (Fig. 4B, Fig. S17 and S18, 2 min), while they appeared completely 'detached' after 10 min of treatment. Of note, the presented methods cannot be used to visualize a direct link between the membrane and nucleoid, but only report on spatial proximity. Interestingly, the loss of nucleoid fine structure appeared already within 2 min, suggesting that entropic effects such as depletion attraction and nucleoid occlusion might play a significant role. Depletion attraction are entropic forces which can cause polymers (here chromosomal DNA) to aggregate or condense in presence of molecular crowders. This effect might cause chromosomal DNA to segregate from the cytosolic phase, which includes such crowders (e.g., ribosomes)[19], and thus localize to the radial cell center. Super-resolved snapshots of nalidixate-treated cells showed that the mid-cell stalled nucleoid remains positioned close to the inner membrane (Fig. 4B).

Release from replication block resulted in structured, strongly elongated nucleoids with membrane-proximal positioning along the entire cell cylinder (Fig. S19). To compare the changes in nucleoid organization, we calculated the RID for all treatments and time points (Fig. S22) and extracted the center of mass of the intensity distribution (Fig. 4C) as well as the distribution width (FWHM) (Fig. 4C). In agreement with CLSM results, rifampicin and chloramphenicol had the strongest effect on nucleoid arrangement with a rapid shift of the DNA distribution toward the radial cell center and a strong decrease in FWHM within the first 10 min. Cell widening upon MP265 and mecillinam exposure led to a shift of DNA signal towards the cell periphery and a narrowing of the radial intensity distribution, both attributed to the reduced effects of 2D projection (Fig. 4C, D, Fig. S22). We also quantified the distance between the nucleoid and the membrane. For untreated cells, we obtained an average DNA-membrane distance of $125 \pm 12$ nm (s.d., $N = 7$ conditions). This value is likely an overestimation of the actual distance, as we had to select certain thresholds for the automated analysis (see methods). Our analysis revealed that the DNA-membrane distance increases during inhibition of transcription and translation, but not during interference with cell wall synthesis (Fig. 4E). In line with our RNL and RID analyses, we observed a maximal DNA-membrane distance at 10 min for both rifampicin ($258 \pm 16$ nm, $N = 84$ cross-sections) and chloramphenicol treatment ($258 \pm 27$ nm, $N = 25$ cross-sections).

Next to the membrane-proximal nucleoid localization, our super-resolved images surprisingly showed qualitative evidence for a moderate association of chromosomal DNA with MreB (Fig. 1B, Fig. 4, Fig. S14–S19). To test whether the apparent spatial correlation between the nucleoid and MreB is random or specific, we developed a circular cross-correlation approach using intensity traces of two channels measured along the cell perimeter (Fig. 4F). In the example cell, the intensity profiles (300 nm line width) of MreB and DNA signals overlap to a large extent. (Fig. 4F, mid panel). Circular shifting of one intensity trace allows calculating the cross-correlation between the two signals, which we performed for lag distances between $-2\,\mu m$ and $2\,\mu m$. A

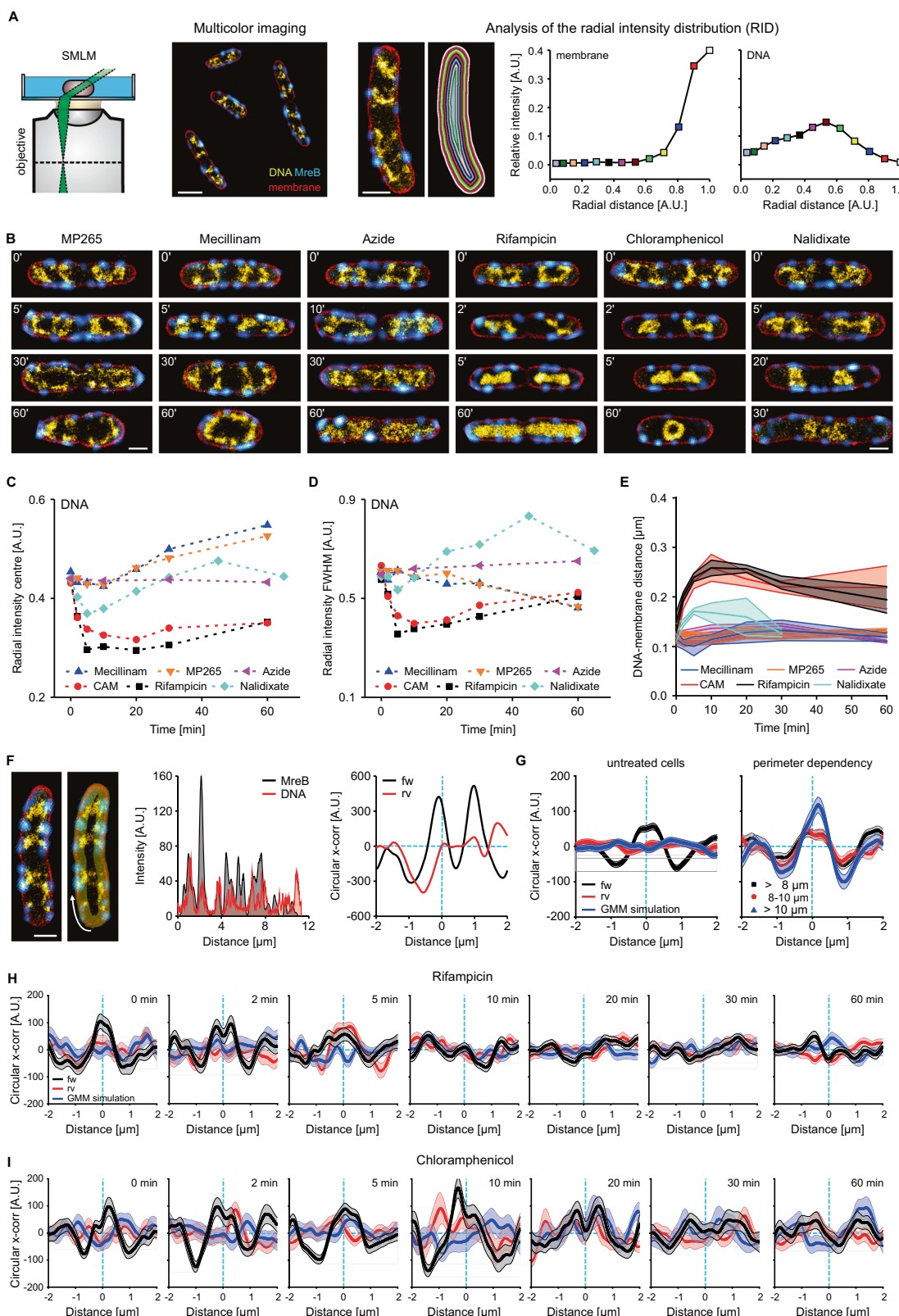

correlation would result in a peak centered at a lag distance of 0, while this peak would be absent in a random intensity distribution. We simulated the latter by two approaches: (i) reversing one intensity trace (rv) and (ii) fitting the MreB intensity distribution with a Gaussian Mixture Model (GMM) and randomizing the trace by shifting the detected peaks. Indeed, we observe a peak for MreB and DNA signal, but not for the reversed DNA intensity trace (Fig. 4F, right panel). We

further validated our method with simulations, in which we equidistantly positioned adjacent foci along a circle (Fig. S23). Rotation of both foci in 5° increments retains cross-correlation of the two signals, while the correlation averages out in the reversed control. Shift of only one channel resulted in a peak shift in the cross-correlation analysis. For MreB signal randomization, we fitted a GMM to individual MreB intensity traces and extracted the number of peaks, peak width and

**Fig. 4 | Super-resolution imaging reveals complete nucleoid reorganization from cell periphery towards the cell center during inhibition of transcription and translation. A** Schematic of multicolor imaging (left) of MreB (cyan), membrane (red) and DNA (yellow hot), as well as RID analysis (right). Cells are segmented into radial slices based on the membrane channel and the relative intensity is determined for each slice. Relative area indicates the distance from cell periphery (1) to the radial cell center (0). An untreated cell is shown as example with slice colors referring to the data points in the RID plot. **B** Representative images of different time points during drug treatment. **C** Plot of RID center of masses vs. drug exposure. Rif = rifampicin, CAM = chloramphenicol. **D** Plot of RID FWHM vs. drug exposure. **E** DNA-membrane distances under different conditions. **F** Circular cross-correlation of MreB and DNA signal reveals non-random proximity. Intensities are measured along the perimeter of the cell (yellow transparent area and white arrow,

left panel). Intensity plots of MreB (black) and DNA (red) of the shown cell (mid panel). Plot of circular cross-correlation (x-corr) with respect to the lag distance. Cross-correlation was calculated for the actual intensity traces (fw, black line) and with one reversed intensity trace (rv, red line) for randomization. Light blue dashed lines indicate the zero values of the x- and y-axis. **G** Averaged circular cross-correlation for all untreated cells (pooled from all treatments, $N = 271$) and different size intervals ($N = 96$, 105, and 78 for increasing perimeters), including reversion- and Gaussian-mixture-model (GMM)-based randomization controls. **H** Averaged circular cross-correlation of rifampicin-treated or **I** chloramphenicol-treated cells. Lines and data points represent mean values and the shaded area the standard errors of the mean. Multiple measurements were performed on a single replicate, chosen from the CLSM experiments shown in Fig. 3. Scale bars are 1 μm. Source data are provided as a Source Data file.

amplitude, followed by random redistribution of the signal along the cell perimeter (Fig. S24, methods). We applied this analysis to published multicolor SMLM data of bacteria overexpressing cytosolic His$_6$-PAmCherry1 or an RpoC-PAmCherry1 fusion protein from the native locus[23,62] (Fig. S25). Cytosolic His$_6$-PAmCherry1 was excluded from the nucleoid region, leading to anti-correlated signal. This anti-correlation was picked up by our circular cross-correlation approach, yielding a negative peak centered at 0. The RpoC-PAmCherry1 fusion, which is known to be mostly associated with the nucleoid in rich medium, resulted in a peak with a positive correlation value, both in untreated and chloramphenicol treated cells. Dissociation from the nucleoid during rifampicin treatment led to a loss of correlation between RpoC and nucleoid signal. No correlation was observed when randomizing the PAmCherry1 intensity traces via the GMM approach (Fig. S25). We further tested the effect of deconvolution on the observed cross-correlation plots (Fig. S26). A peak centered around a shift distance of 0 was observed in images with raw MreB-signal and signal deconvolved either by the Wiener Filter Preconditioned Landweber or Richardson-Lucy algorithms.

Together, we thus assume that this method is suitable for detecting non-random association of two signals, in this case the *E. coli* nucleoid and elongasomes. Applying the method to the super-resolved images, the average circular cross-correlation showed a local maximum at the center position for untreated cultures (Fig. 4G, $N = 271$). This indicates that the observed spatial correlation between the nucleoid and MreB are non-random across cell sizes (Fig. 4F, right panel). Of note, the peak heights in the cross-correlation analyses between MreB and DNA signals (~50–100 A.U.) are smaller compared to the RpoC controls (Figs. 4G, S25) (~400–500 A.U.), indicating that the DNA-elongasome correlation is less prominent. However, a quantitative comparison is challenging, as our approach uses z-normalized intensity traces to enable cross-correlation analysis on data with varying signal strengths. This results in a correlation with arbitrary units instead of classical correlation coefficient, better suited to study the spatial context of the signals' correlation instead of absolute colocalization. Inhibition of transcription and translation lead to the loss of correlation (Fig. 4H, I), indicating that protein biosynthesis positions the nucleoid close to the inner membrane. This loss is gradual, as indicated by the reduction in peak amplitude during rifampicin treatment (Fig. 4H). When interfering with cell wall synthesis, we detected the strong reorganization of MreB at early time points of MP265 treatment as an anticorrelation in the average cross-correlation plot. This was not the case for Mecillinam-treated cells (Fig. S27). Signals for nalidixate-treated cells remain correlated, while correlation is lost during azide treatment due to MreB reorganization.

## Altered RNase E localization or expression level do not cause major changes in nucleoid morphology

Transertion requires translated mRNAs to be in close proximity to the cell membrane to establish the indirect nucleoid-membrane links.

Such a membrane localization was found for mRNAs that encode proteins which are co-translationally inserted via the signal recognition particle (SRP) pathway[63]. These mRNAs exhibit a reduced lifetime, likely due to the spatial proximity to the membrane-associated RNA degradasome. Deletion of the membrane-targeting A-segment of RNase E, the main component of the RNA degradasome, increased mRNA stability. We thus sought to test whether RNase E localization has an effect on nucleoid morphology and performed super-resolution imaging of bacteria expressing wild type RNase E and RNase E deleted for the A-segment (ΔA)[64]. While these measurements did not reveal a dramatic reorganization of the nucleoid, they showed a slight increase of the DNA-membrane distance (Fig. 5A, B) from 94 ± 4 nm ($N = 78$ cells from three replicates) to 112 ± 2 nm ($N = 81$ cells from three replicates) ($p = 0.0019$, unpaired Welch's $t$-test), which might be a consequence of the reduced growth rate ($t_d = 29$ min for the WT and 38 min for the ΔA mutant). We also overexpressed RNase E–YFP from an inducible plasmid in a Δrne deletion strain, resulting in a - 4.1-fold difference in RNase E signal intensity, but showing no correlation between the DNA-membrane distance and RNase E intensity (Fig. 5C, D, $p = 0.144$, unpaired Welch's $t$-test).

## Nucleoid compaction depends on active transcription, but not on cell geometry

Recent work with wall-less L-forms of *B. subtilis* showed that nucleoid positioning depends on confinement and cell geometry[15]. To test whether this is also the case for nucleoid compaction in *E. coli*, we inhibited protein biosynthesis in cells that had been widened by pre-treatment with MP265 using rifampicin or chloramphenicol (Figs. 6, S28). As expected, cell widening reduced the effect of 2D projection, as larger parts of the nucleoid are positioned outside of the projected volume (see Supplementary Note 1). This leads to a shift of the RID center of mass (CoM) from 0.41 A.U. in untreated cells ($N = 39$) to 0.53 A.U. in widened cells ($N = 66$). Interestingly, inhibition of transcription initiation in pre-widened cells using rifampicin results in a detached nucleoid that maintains visible sub-structuring (Fig. 6A). This stands in contrast to elongated cells, where transcription inhibition resulted in a relatively unstructured and elongated nucleoid positioned along the bacterial long axis (Fig. 4B). In pre-widened cells, the RID CoM dropped from 0.53 A.U. (MP265 only) to 0.37 A.U. (MP265 + 5 min rifampicin, $N = 57$) and 0.39 A.U. (MP265 + 10 min rifampicin, $N = 53$), while it dropped to 0.30 A.U. for both 5 and 10 min rifampicin treatment in elongated cells ($N = 45$ and 58, respectively). Inhibition of translation with chloramphenicol resulted in a condensed nucleoid as it is also observed in elongated cells (Fig. 4B). However, the positioning at the cell center is much more apparent in widened cells as also shown by the shift of the nucleoid signal towards the cell center in the RID analysis (Fig. 6B) (CoM = 0.36 A.U., $N = 57$, and 0.32 A.U., $N = 52$ for 10 and 20 min chloramphenicol, respectively). To determine whether the relative nucleoid size changes with morphology, we segmented super-resolved nucleoids using a neural network that we trained on manually segmented nucleoids (see methods). The analysis revealed that the

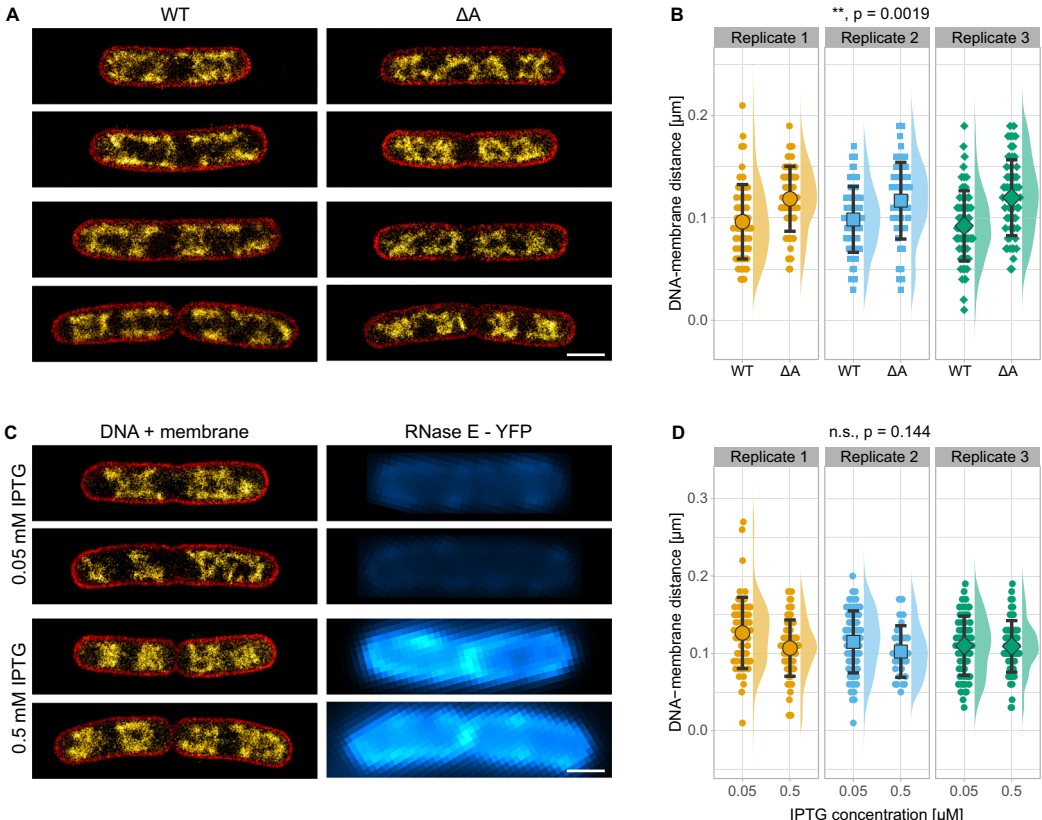

**Fig. 5 | Effect of RNase E localization and expression level on *E. coli* nucleoid organization. A** Super-resolution imaging of *E. coli* cells expressing WT RNase E or the cytosolic ΔA mutant, labeled for membrane (red) and DNA (yellow hot). **B** DNA-membrane distance of the strains shown in (**A**). An average distance of $96 \pm 4$ nm (s.d., 3 biological replicates, $N_{cells} = 24$, 27 and 27, $N_{sections} = 62$, 61, and 69) was observed for the strain expressing WT RNase E and $119 \pm 2$ nm (s.d., 3 biological replicates, $N_{cells} = 26$, 20, and 35, $N_{sections} = 56$, 52, and 60) for the strain expressing the ΔA mutant. $p = 0.0019$ (two-sided, unpaired Welch's *t*-test). **C** Super-resolution imaging of *E. coli* cells with varying RNase E–YFP levels (cyan), labeled for

membrane (red) and DNA (yellow hot). **D** Correlation of DNA-membrane distance and RNase E expression level. Expression was of plasmid-borne RNase E–YFP was controlled by IPTG concentration (0.05 and 0.5 mM). An average distance of $117 \pm 9$ nm (s.d., 3 biological replicates, $N_{cells} = 20$, 31, and 24, $N_{sections} = 59$, 72, and 63) was observed for induction of RNaseE-YFP expression with 0.05 mM IPTG and $104 \pm 6$ nm (s.d., 3 biological replicates, $N_{cells} = 20$, 5, and 20, $N_{sections} = 64$, 29, and 63) for induction with 0.5 mM IPTG. $p = 0.144$ (two-sided, unpaired Welch's *t*-test). Distributions in (**B**, **D**) show mean values and standard deviations. Scale bars are 1 μm. Source data are provided as a Source Data file.

relative nucleoid size in rifampicin- and chloramphenicol-treated bacteria is independent of cell geometry (Fig. 5C).

This leads to a model in which nucleoid positioning depends on protein biosynthesis and entropic forces, while nucleoid compaction is mainly caused by active transcription (Fig. 6D)[54]. Nucleoid size hereby seems to depend on the cell volume, as approximated by the relative nucleoid size obtained in 2D projections. To test whether cellular dimensions affect nucleoid condensation, we recorded dual-color PAINT images of the rod-shaped Gram-negative entomopathogenic bacterium *Xenorhabdus doucetiae*. We chose this organism because it exhibits larger cellular dimensions, in particular an increased cell diameter ($1.42 \pm 0.12$ μm), compared to *E. coli*[65], while other features such as chromosome size, growth rate and presence of nucleoid associated proteins are comparable (Supplementary Note 3). In contrast to *E. coli*, the nucleoid of *X. doucetiae* cells populated the entire bacterial long axis (Fig. 7, untreated) and showed a higher relative nucleoid size ($0.42 \pm 0.08$ vs. $0.26 \pm 0.06$) (Fig. S28). However, it also detaches from the inner membrane upon rifampicin treatment, revealing a structured nucleoid similar to widened *E. coli* cells (Fig. 5A). Detached, but fully replicated sister chromosomes hereby remain segregated, highlighting the role of entropic forces in chromosome organization[12].

## Discussion

In this study, we investigated the nucleoid organization in rapidly growing *E. coli* cells during unperturbed and perturbed growth. We

used confocal microscopy to visualize the global effect of various antibiotics on cell shape and nucleoid morphology and performed super-resolution microscopy measurements to provide highly resolved snapshots of drug-treated cells.

We found that the *E. coli* nucleoid is positioned in close proximity to the inner membrane during unperturbed growth in rich medium, while it occupies the radial cell center during slow growth (Fig. 1). 3D single-molecule localization microscopy hereby revealed a membrane-proximal nucleoid arrangement with a strikingly clear DNA-free region in the radial cell center (Supplementary Movie 1) that likely harbors the cytosolic phase, including proteins (Fig. S25), ribosomes[16], RNAs and other biomolecules. We verified this arrangement in vivo by imaging vertically aligned cells with VerCINI (Fig. 1C)[50,51]. A twisted, membrane-proximal arrangement of condensed DNA was also observed in or suggested by other studies[6,9,10,47,48]. Fisher and colleagues hypothesized that the nucleoid represents a rigid filament in a too small cell cylinder, thus being forced into a twisted configuration[10]. However, the strong and rapid reorganization observed during inhibition of protein biosynthesis (Fig. 3) does not support this hypothesis. Another study suggests that a condensed, donut-shaped nucleoid is thermodynamically favorable, but it does not take biosynthetic processes into consideration[7]. Recently, a computational study showed that attractive and repulsive interactions between ribosomes and the nucleoid can cause DNA depletion at the radial cell center, fitting the observation we made in our experiments[66]. Surprisingly, we only observed this

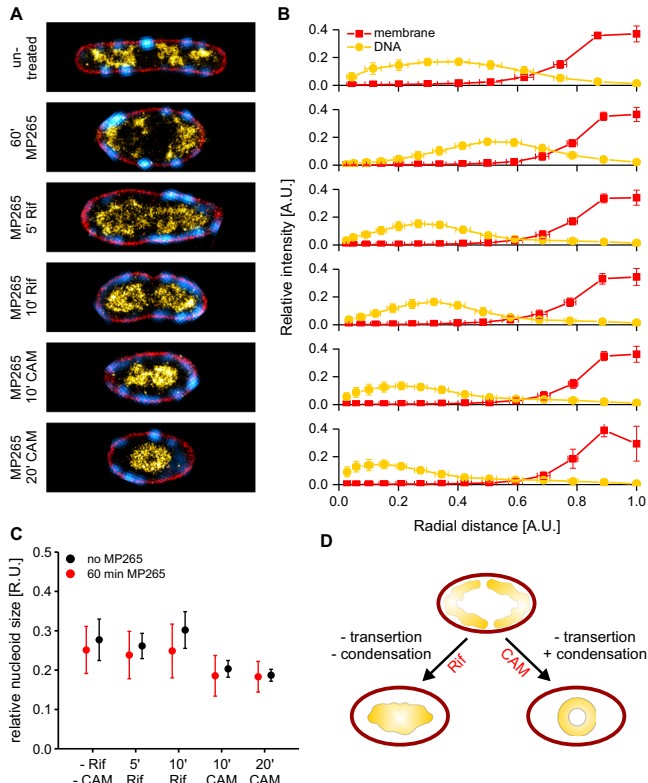

**Fig. 6 | Inhibition of transcription and translation in widened cells.**
**A** Representative cells for the different conditions labeled for MreB$^{sw}$-sfGFP (cyan), membrane (red) and DNA (yellow hot). **B** Average radial intensity distributions. Cell counts are provided in Table S5. Cells were extracted from multiple measurements of the same experiment. **C** Relative nucleoid size of drug treatments in normal (no MP265, Fig. 4B) and widened cells (60 min MP265). Rif = rifampicin, CAM = chloramphenicol. $N_{cells}$ = 36 and 39 (-Rif/CAM, normal and widened), 45 and 56 (5 min Rif, normal and widened), 58 and 73 (10 min Rif, normal and widened), 19 and 70 (10 min CAM, normal and widened), 17 and 58 (10 min CAM, normal and widened). Data points represent mean values and error bars the standard deviations. **D** Schematic of transcription/translation inhibition in widened cells. Source data are provided as a Source Data file.

phenotype during rapid growth in LB ($t_d$ = 27 min), but not in slowly growing cells (M9, $t_d$ = 120 min), where the nucleoid positions at the radial cell center. This could be the result of an increased rate of transertion during fast growth. Although the relative abundance of periplasmic and outer membrane proteins decreases with increasing growth rate, the relative amount of inner membrane proteins was found to remain constant[67]. To sustain this constant level during rapid growth, the expression dynamics of inner membrane proteins must be faster. The high concentration of ribosomes would support such a model, in particular as ribosomes were found to show an increased formation of polysomes during fast growth[68]. Furthermore, mRNA turnover is elevated during fast growth[69], eventually increasing the rate of transertion in comparison to slow growth, where increased mRNA stability favors translation of completed transcripts that are detached from DNA.

Our drug-treatment data supports a model where protein biosynthesis, likely via transertion, couples the nucleoid to the inner membrane. This strengthens observations of previous work[19] and provides high-resolution data of nucleoid reorganization dynamics. The rapid detachment upon rifampicin and chloramphenicol treatment occurs within the first 10 minutes, a time scale that matches transcription and translation of most genes[19]. The gradual detachment observed in early time points might thus reflect the runoff of active transcription/translation events (Figs. 4, S17, and S18). As mRNA

lifetimes in rapidly growing bacteria are at the minute range[63], total cellular mRNA levels decrease during rifampicin exposure. Conversely, chloramphenicol treatment was found to stabilize mRNAs and increase rRNA synthesis, resulting in elevated cellular RNA levels[70]. Solvent quality, which is highly affected by RNA[17], should thus differ significantly between chloramphenicol and rifampicin treated cells. However, we observed almost identical nucleoid phenotypes during the first 10 min of drug exposure, arguing against changes in solvent quality as driver of this initial reorganization.

The different responses of the two treatments at longer drug exposure (Figs. 3 and 4) could be explained by the condensing force of active transcription, which is present during chloramphenicol but not rifampicin treatment[54], or a varying amount of supercoiling, which is enhanced by transcription[71]. Although we think that changes in solvent quality have no or limited effect at short drug exposure, it still might affect nucleoid compaction at longer treatment durations, when differences in RNA levels are largest. Notably, the observed changes in nucleoid positioning are almost binary, showing the entire DNA in membrane-proximity in widened cells and a complete repositioning of the nucleoid towards the cell center upon inhibition of transcription and translation (Figs. 6 and S28). This dramatic effect of transcription on nucleoid structure agrees well with a recent study that identified transcription as an elementary regulator of chromosome structuring[30].

Our measurements do not exclude that DNA-membrane contacts occur independently of transertion. Especially for long exposure times of rifampicin and chloramphenicol, DNA-membrane distances decrease, raising the question whether DNA-membrane links are re-established (Figs. 4 and S17, 18). Work from Weber and colleagues revealed that the mobility of genetic loci strongly increases during rifampicin treatment, while only showing modest increase during chloramphenicol treatment[29]. We thus conclude that the nucleoid in rifampicin-treated cells is in a decondensed and highly mobile state, in which the proximity to the membrane is of random nature. On the other hand, the lower mobility in untreated cells could be caused by nucleoid attachment, which introduces anchor points that restrict DNA movement.

One requirement of transertion is the positioning of mRNA close to the membrane[63,72]. mRNAs encoding membrane proteins, which are inserted via the SRP-pathway, specifically localize at the inner membrane and this only if they are actively transcribed. As a significant fraction of transcription occurs co-translationally, transertion-mediated positioning of the nucleoid at the membrane is well possible[63]. It was further found that RNase E localization affects mRNA stability, with cytosolic localization of a ΔA RNase E mutant leading to increased RNA stability, particularly for membrane-protein-encoding RNAs. As RNase E exhibits an increased activity on operons and mRNAs encoding membrane and periplasmic proteins, we would expect increased RNase E expression levels to cause rapid degradation of such transcripts, causing a relocation of the nucleoid towards the cell center. Induction of plasmid-encoded RNase E expression with varying inducer concentrations resulted in a 4.1-fold difference in copy number. However, our measurements did not reveal major differences in nucleoid organization for both altered RNase E localization and abundance (Fig. 5), indicating that degradasome activity, at least under the condition tested, does not significant contribute to nucleoid organization. It is also possible that mRNAs are (partially) protected against degradation and are thus unaffected by elevated RNase E levels. Another possibility is that cells increase transcription rates to compensate for increased degradation, maintaining nucleoid positioning close to the membrane.

Surprisingly, we found evidence of colocalization between MreB and the nucleoid. This could be mediated by previously reported interactions of MreB with EF-Tu, RNA polymerase and SecA, all representing potential links between elongasomes and the transertion machinery[61,73–75]. However, our data does not support an essential

no treatment

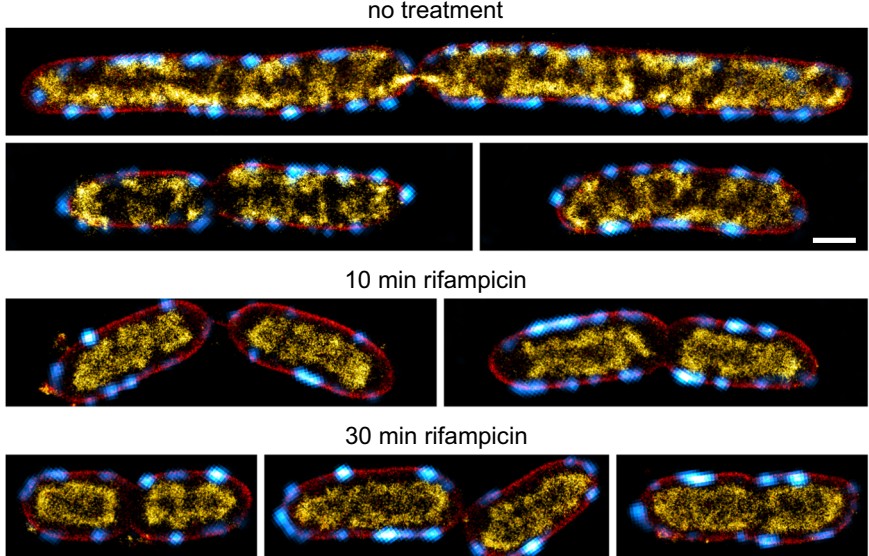

10 min rifampicin

30 min rifampicin

**Fig. 7 | Nucleoid organization in the insect pathogen *Xenorhabdus doucetiae*.** Deconvolved signal of MreB[sw]-sfGFP is shown in cyan, while super-resolved PAINT images of membranes and DNA are shown in red and yellow hot, respectively. Scale bar is 1 μm.

direct role of MreB in nucleoid anchoring, as MreB depolymerization does not affect nucleoid distribution and chromosomal DNA is not recruited to the cell poles upon MreB aggregation (Figs. 2B and 3B). Instead, we observe a strong effect of cell morphology on nucleoid distribution. Both during MP265 and mecillinam treatment, nucleoid reorganization coincides with rod-to-sphere transition (Figs. 2, 3B, and Fig. 4B). In widened cells, chromosomal DNA is also found at polar membranes, a phenomenon that is typically not observed in rod-shaped cells (Fig. 4). This indicates that confinement and/or cell curvature are involved in nucleoid positioning, a hypothesis that is line with literature[7,12,13]. While cell widening did not change the relative nucleoid size both during unperturbed and perturbed protein biosynthesis, it had an influence on the expansion of center-positioned nucleoids (Fig. 6). Cells treated with MP265 and rifampicin showed nucleoid sub-structuring that is not visible in rod-shaped cells, in which the nucleoid shows a sausage-like arrangement along the bacterial long axis (Fig. 4). We attribute this observation to changes in confinement along the cells' short axis as was recently observed in L-form bacteria confined in channels of various sizes[15]. Increased mixing of ribosomes and the nucleoid in a larger space might also promote nucleoid expansion, but this remains to be tested by correlative imaging of ribosomes and DNA and/or single-particle tracking. A more expanded nucleoid was also observed in larger *Xenorhabdus doucetiae* cells (Fig. 7), supporting the hypothesis that cellular dimensions and thus confinement affects nucleoid organization.

We further observed that replication arrest by nalidixate resulted in a mid-cell-positioned nucleoid that still showed membrane-proximal localization (Figs. 4 and S19). Continued cell growth led to large nucleoid-free regions that are rapidly repopulated by DNA upon removal of nalidixate (Figs. 3E and S19). The dynamics hereby exceeded the speed of cell elongation, thus suggesting entropic forces as the main cause of nucleoid expansion. As shown by Jun and Mulder[12], replicating chromosomes might segregate rapidly due to repelling forces acting between distinct topological DNA domains (as visible e.g., in MP265/rifampicin double-treated cells, Fig. 6), thus maximizing the conformational entropy of the system. This is in agreement with previous work that showed nucleoid expansion in elongating but non-replicating cells as well as rapid nucleoid repositioning upon asymmetric cell division in elongated cells[13,15]. Next to changes in nucleoid organization, our analyses also revealed a rapid reorganization of

MreB to the cell poles upon inhibition of polymerization. Work by Kawazura et al. showed that MreB monomers are recruited to anionic lipids, which are enriched at bacterial cell poles[58]. However, we also observed subtle MreB recruitment to the cell poles during inhibition of peptidoglycan crosslinking (Figs. 2 and 3B, D) at a similar time scale. This indicates that crosslinking activity might affect MreB polymerization.

In conclusion, our super-resolution imaging of drug-treated *E. coli* cells provides high-resolution snapshots revealing a condensed, membrane-proximal configuration of the nucleoid during rapid growth. The dramatic reorganization of the nucleoid upon inhibition of protein biosynthesis, occurring within minutes, indicates that transertion might be required to maintain this expanded, membrane-proximal state. While our data cannot conclusively demonstrate a direct role for transertion in shaping nucleoid morphology, it provides compelling evidence for transertion as a key organizing principle. By tracking the rapid response of the nucleoid to potential disruption of transertion, we have captured valuable insights into the dynamics of this process. Going forward, correlative imaging of the transertion machinery components and the nucleoid will help elucidate the molecular mechanisms underlying this phenomenon. Beyond elucidating the role of transertion, our work establishes analytical tools to quantitatively assess nucleoid organization relative to cellular landmarks like the membrane and proteins. The automated image analysis routines we developed, including population averaging, RID analysis and circular cross-correlation, can provide versatile new approaches for investigating spatiotemporal organization in bacteria. In summary, our high-resolution snapshots of antibiotic-treated cells reveal rapid dynamics of transertion-mediated effects on nucleoid morphology, while also demonstrating broadly applicable strategies for relating subcellular structure to function.

## Methods

### Bacterial strains and culturing

Bacterial strains used in this study are listed in Table S1. ON cultures were inoculated from single colonies into LB Lennox (5 g NaCl) and grown at 32 °C shaking at 200 rpm. The next day, cultures were diluted 1:200 into fresh LB and grown to exponential phase incubated at 32 °C and 200 rpm. $OD_{600}$ was checked every 30 min to ensure proper growth and to determine the culture mass doubling time. The following antibiotics/compounds were added at an $OD_{600}$

of $0.25 \pm 0.2$: MP265 (25 μM), mecillinam (2 μg/ml), rifampicin (100 μg/ml), chloramphenicol (50 μg/ml), sodium azide (1 mM), and nalidixic acid (50 μg/ml). Antibiotic stock solutions were prepared freshly before use. In combinatorial drug experiments, 25 μM MP265 was added at $OD_{600} \sim 0.25$, and rifampicin or chloramphenicol were added for the indicated duration at time points that sum up to 60 min total MP265 treatment. *Xenorhabdus doucetiae* was grown in LB Lennox at 30 °C and 200 rpm. Similar to *E. coli*, rifampicin (100 μg/ml) was added during mid exponential phase. The strain used for experiments of varying RNase E abundance (SLP60) was inoculated from single colonies into LB Lennox including 0.1 mM IPTG and grown overnight at 37 °C. Cells were then back-diluted 1000-fold into LB with 0.05 mM or 0.5 mM IPTG and grown to OD600 ~ 0.5–0.7 before chemical fixation.

## Strain construction

*E. coli* strain CS1 (*E. coli* MG1655 chromosomally expressing MreB$^{sw}$-sfGFP and HupA-mScarlet-I) was constructed using λ-red recombineering. It carries a fluorescent protein fusion of the cytoskeletal protein MreB and msfGFP (parental strain NO34[49]) and a fluorescent protein fusion of the nucleoid-associated protein HU-alpha and mScarlet-I. For recombineering, plasmid pKD46 was transformed into electro-competent NO34. The DNA fragment for C-terminal insertion of mScarlet-I including a flexible linker (GSAGSAAGSGEF) and a chloramphenicol resistance cassette was generated as follows: Fragment 1 contains a 147 bp overlap to the C-terminal sequence of *hupA* and the linker. It was amplified from the NO34 genome using primers CS_FFM_002 and CS_FFM_003. Fragment 2 (linker-mScarlet-I) was amplified from Addgene plasmid #85044 (pmScarlet-I_C1[76]) using primers CS_FFM_005 and CS_FFM_006. Fragment 3 including the Chloramphenicol resistance cassette and a 59 bp overlap to the downstream region of *hupA* (mScarlet-I-FRT-CAT-FRT-hupA_ds) was amplified from Addgene plasmid #101148 (pmMaple3-CAM[77]) using primers CS_FFM_017 and CS_FFM_018. Fragments 1 and 2 were fused in a 2-step PCR using amplification primers CS_FFM_002 and CS_FFM_006, and the resulting fragment was fused similarly with fragment 3 using amplification primers CS_FFM_002 and CS_FFM_019. The resulting DNA fragment (1857 bp) was electroporated into NO34 carrying pKD46. Clones carrying the insertion were selected on plates containing Chloramphenicol and clones were verified by sequencing and fluorescence microscopy.

*E. coli* strain CS2 (*E. coli* MG1655 chromosomally expressing MreB$^{sw}$-sfGFP and H-NS-mScarlet-I) was constructed similar to CS1. It carries a fluorescent protein fusion of the cytoskeletal protein MreB and msfGFP (parental strain NO34[49]) and a fluorescent protein fusion of the nucleoid-associated protein H-NS and mScarlet-I. The DNA fragment for C-terminal insertion of mScarlet-I including a flexible linker (GSAGSAAGSGEF) and a chloramphenicol resistance cassette was generated as follows: Fragment 1 contains a 155 bp overlap to the C-terminal sequence of *hns* and the linker. It was amplified from the NO34 genome using primers CS_FFM_012 and CS_FFM_013. Fragment 2 is identical to the fragment used for construction of CS1. Fragment 3 including the Chloramphenicol resistance cassette and a 54 bp overlap to the downstream region of *hns* (mScarlet-I-FRT-CAT-FRT-hns_ds) was amplified from Addgene plasmid #101148 (pmMaple3-CAM) using primers CS_FFM_017 and CS_FFM_020. Fragments 1 and 2 were fused in a 2-step PCR using amplification primers CS_FFM_012 and CS_FFM_006, and the resulting fragment was fused similarly with fragment 3 using amplification primers CS_FFM_012 and CS_FFM_021. The resulting DNA fragment (1860 bp) was electroporated into NO34 carrying pKD46. Clones carrying the insertion were selected on plates containing Chloramphenicol and clones were verified by sequencing and fluorescence microscopy.

*Xenorhabdus doucetiae* strain CS_Xd1 was constructed using a pDS132-based suicide plasmid. First, a pCK-MreB-sfGFP plasmid

(pDS132 with chloramphenicol resistance cassette) was constructed. The plasmid contains the *X. doucetiae* MreB gene with an msfGFP inserted into an internal loop between amino acids T228 and D229. For the construction of the plasmid, Fragment 1 (left homology region of MreB) was amplified using primers CS_MPI_003 and CS_MPI_004. Fragment 2 is the msfGFP sequence (*E. coli* codon-optimized) flanked by two linker regions ('SGSS' on the left and 'SGAPG'' on the right) and was amplified from genomic DNA of the *E. coli* strain NO34 using primers CS_MPI_018 and CS_MPI_019. Fragment 3 contains the remaining sequence of the MrebCD operon and was amplified with primers CS_MPI_005 and CS_MPI_006. As backbone, plasmid pCK-CipA (suicide vector pDS132 derivative with Chlorampenicol resistance cassette) was digested with PstI and BglII (NEB). Fragments and backbone were assembled into one plasmid using Gibson Assembly (NEB HiFi DNA assembly kit) and electroporated into competent cells of the *E. coli* ST17 λ-pir conjugation strain. Clones were selected on chloramphenicol-containing agar plates and verified by colony PCR using primers VpDS132_fw and VpDS132_rv. The plasmid was transferred into *X. doucetiae* WT cells via conjugation. An *X. doucetiae* clone with chromosomally integrated pDS132 plasmid was grown overnight without antibiotics and 5 μl were streaked on LB agar plates containing 6% sucrose for the second recombination. Clones were screened by fluorescence and insertion of GFP at the proper site was confirmed by sequencing. PCR templates for sequencing were amplified using primers CS_MPI_020 and CS_MPI_021.

Primers used for cloning are listed in Table S2. Strains double labeled for MreB$^{sw}$-sfGFP and HU-α-mScarlet-I or H-NS-mScarlet-I were generated using lambda RED recombineering[78]. mScarlet-I was amplified from plasmid pmScarlet-I_C1 (Addgene # #85044[76]) and the chloramphenicol resistance cassette from plasmid pmMaple3-CAM (Addgene #101148[77]). Strains were verified by sequencing and fluorescence microscopy. *Xenorhabdus doucetiae* MreB$^{sw}$-sfGFP strain was constructed using a pDS132-based suicide plasmid[79] and the amino acid linkers (SGSS-msfGFP-SGAP) as described in Ouzounov et al.[49]. Due to differences in the amino acid sequence of MreB, SGSS-msfGFP-SGAP was inserted between T228 and D229 instead of G228 and D229.

## Vertical cell imaging by nanostructured immobilization (VerCINI)

Overnight cultures were set up from a single colony in 2 ml M9 + glucose or LB and incubated at 37 °C with orbital agitation at 175 rpm. The following morning, cultures were diluted to an $OD_{600}$ between 0.05 and 0.1 in 5 ml volumes then incubated at 30 or 37 °C with orbital agitation at 175 rpm.

A VerCINI agarose pad was prepared by spotting 6% molten UltraPure agarose (VWR) dissolved in media onto a micropillar wafer and transferred into a Geneframe (Thermo Fisher Scientific) mounted on a glass slide as previously described[51]. When cultures reached mid-exponential phase, at an $OD_{600}$ between 0.6 and 0.8, 500 μl of culture was centrifuged at 16,900 *g* for 1 min then resuspended in 10 μl pre-warmed media. The 10 μl concentrated culture was spotted onto a pre-warmed VerCINI agarose pad. The slide was then centrifuged at 3220 *g* for 4 min. The sample was then washed with pre-warmed media, to remove horizontal cells from the pad and air-dried, before a cover slip (VWR, 22 × 22 mm$^2$, thickness no. 1.5) was applied.

On a custom single-molecule microscope, samples were illuminated with 488 and 561 nm lasers (Obis) with a power density of ~17 W/cm$^2$ for 500 ms with HILO illumination[80] achieved by galvanometer-driven mirrors (Thorlabs) rotating at 200 Hz. A 100x TIRF objective (Nikon CFI Apochromat TIRF 100XC Oil), a 200 mm tube lens (Thorlabs TTL200) and a Prime BSI sCMOS camera (Teledyne Photometrics) were used for imaging, giving effective image

pixel size of 65 nm/pixel. Images were denoised in Fiji[81] using the PureDenoise plugin[82].

## Chemical fixation and immobilization

Chemical fixation and immobilization was performed as described previously[23]. Aliquots were taken at defined time points of drug treatment and fixed by directly adding fixation solution, providing a final concentration of 2% methanol-free formaldehyde (Thermo Fisher Scientific) and 0.2% EM-grade glutaraldehyde (Electron Microscopy Sciences) in 33 mM $NaPO_4$ buffer (pH 7.5). Cells were fixed for 12 min at room temperature (RT) and quenched by replacing the fixation solution with freshly made 0.2% $NaBH_4$ (Sigma Aldrich) in PBS for 3 min. Afterwards, cells were washed thrice with PBS and immobilized on KOH-cleaned (3 M KOH for 1 h), poly-L-lysine coated 8-well chamberslides (Sarstedt GmbH). All centrifugation steps were performed for 2 min at $6000\,g$ in a benchtop centrifuge (Thermo Fisher Scientific). For PAINT imaging, cells were permeabilized 30 min with 0.5% Triton-X-100 (Sigma Aldrich) and washed twice with PBS. 60 nm and 80 nm gold beads (Nanopartz Inc.) were added as fiducial markers at a concentration of 1:200 v/v each and allowed to settle for 30 min. Excess gold beads were removed by washing the well twice with PBS.

## Click-labeling of metabolically labeled DNA

For the 3D measurement shown in Fig. 1A, cultures were grown as specified before except for adding 5-ethynyl-2-deoxyuridine (EdU, Baseclick GmbH) to the culture at $OD_{600} = 0.25$ for 30 min. Cells were then fixed and immobilized as described above. Coupling with Alexa Fluor 647 azide (Thermo Fisher Scientific) was performed via copper-catalyzed click chemistry as described elsewhere[45]. After the click reaction, cells were washed thoroughly with PBS and subjected to dSTORM imaging.

## Confocal imaging of fixed cells

Immobilized cells were stained for DNA using 600 nM DAPI for 10 min. 100 nM Nile Red was added to the cells. As Nile Red is fluorogenic, samples were imaged without the removal of the staining solution. CLSM imaging was performed on a commercial LSM710 microscope (Zeiss, Germany) bearing a Plan-Apo 63x oil objective (1.4 NA). The internal 543 and 633 nm laser lines and an external LGN3001 argon-ion laser (LASOS Lasertechnik GmbH, Germany) were used as excitation light sources. A 405-100 C Coherent Cube diode laser (Coherent GmbH, Germany) was additionally coupled into the microscope via an optical fiber. Suitable filter sets were chosen for each fluorescent probe. Triple-color measurements of NR-stained membrane, the GFP fusion protein and DAPI-stained chromosomal DNA were performed in sequential imaging mode following the indicated order to minimize photobleaching of GFP. Imaging was carried out using a pixel size of 84 nm, 16-bit image depth, 16.2 µs pixel dwell time, 2x line averaging and 1 AU pinhole size. NR, GFP and DAPI were excited using 543, 488, and 405 nm illumination, respectively. Fluorescent signal was detected at gains optimized for signal-to-noise ratio of each fluorophore. Emission detection windows were set to 547–753 nm, 492–541 nm and 409–542 nm for NR, GFP, and DAPI, respectively in order to minimize crosstalk.

## Live-cell imaging

Live-cell imaging was performed on a commercial Leica SP8 confocal microscopy (Leica Microsystems GmbH), equipped with a white-light laser and one hybrid detector. Cells were immobilized on agar pads poured into gene frames[83] and imaged at room temperature. Dual-color imaging was performed in line sequential mode to reduce displacements caused by sample drift and the dynamic nature of the structures imaged. Obtained time series were denoised using PureDenoise[82] and corrected for photobleaching using the Fiji plugin "Bleach Correction" with histogram normalization.

## Single-molecule localization microscopy

2D and 3D SMLM experiments were carried out on a custom build setup for single molecule detection or a commercial Nikon N-STORM system (Nikon Instruments). The custom-built system consists of a Nikon Ti-Eclipse body mounted with an 100x Apo TIRF oil objective (NA 1.49, Nikon Instruments), a perfect focus system (Ti-PFS; Nikon), a MCL Nano-Drive piezo stage (Mad City Labs), an adjustable TIRF mirror and a custom cylindrical lens for 3D imaging (RCX-39.0.38.0-5000.0-C-425-675, 10 m focal length, CVI Laser Optics, UK). An Innova 70 C Spectrum laser (Coherent) and a 405 nm UV diode laser (Coherent CubeTM 405-100 C, Coherent) were used as excitation sources and laser lines were selected using an AOTFnC-VIS-TN acousto-optical tunable filter (AOTF, AA Opto Electronic). For dual-color PAINT imaging, 200–400 pM $JF_{646}$-Hoechst (DNA) and Nile Red (membrane) were added to the well in 150 mM tris pH 8.0[23]. Fluorophores were excited with 1–3 kW/cm² 561 and 647 nm laser light and image sequences were recorded in oblique illumination at frame rates between 33 and 50 Hz. 3D dSTORM imaging of click-labeled DNA (Fig. 1) was performed in PBS containing 100 mM MEA (Sigma Aldrich), adjusted to pH 8.5 using NaOH.

## Pseudo-3-color imaging

To co-image the nucleoid and membrane together with MreB and without chromatic aberrations (Fig. 1B), we imaged NO34 cells together with $JF_{503}$-Hoechst and Potomac Gold. Using a filter set that shows some crosstalk of these dyes in the 488 nm channel, DNA and subtle membrane signals are visible in the reconstructed image. MreB stacks were initially acquired using low laser intensity and subsequently bleached to facilitate single-molecule imaging of the transiently binding fluorophores.

## Analysis of PAINT data

PAINT images of DNA and membranes were analyzed using rapid-STORM v3.31[84]. Binding events were fitted using a free parameter Gaussian fit with a threshold of 200 ($JF_{646}$-Hoechst) or 50 photons (Nile Red). Filtering was performed according to PSF width ($JF_{646}$-Hoechst: 240 nm $<FWHM_{x/y} < 520$ nm; Nile Red: 220 nm $<FWHM_{x/y} < 440$ nm) and symmetry (FHWM ratios between 0.7 and 1.3). Subsequent frame localizations were grouped into single localizations. Chromatic aberrations between channels were corrected using linear alignment matrices obtained from calibration images of fiducial markers, with the MreB channel serving as reference channel. Data for *Xenorhabdus doucetiae* was processed with Picasso[85] using a similar routine.

## Generation of diffraction limited images

Due to the low concentration of bound labels in PAINT imaging, we generated diffraction limited images from the SMLM time series. For this, we calculated the standard deviation image of 5000 frames for Nile Red and JF646-Hoechst measurements. Standard deviation images provide a higher contrast and a slightly improved resolution compared to average images.

## Deconvolution of widefield MreB images

For deconvolution of the MreB channel, stacks with 100 nm spacing were recorded. Stacks were deconvolved with an experimental PSF obtained from fiducial markers (100 nm TetraSpeck microspheres, Thermo Fisher Scientific) using Macro M1. All macros used in this study are provided as Supplementary Software and listed in Table S3. Multiple PSFs were extracted from the stack and averaged using a custom-written Fiji macro. Prior to deconvolution, MreB^sw-sfGFP stacks and the experimental PSF were scaled by a factor of 2 using bicubic interpolation. Stacks were deconvolved using the Fiji plugin 'Parallel Iterative Deconvolution" using the Wiener Filter Preconditioned Landweber algorithm. Control deconvolution with the Richardson-

Lucy algorithm was perfomed on non-scaled images (Fig. S27) using DeconvolutionLab2[86].

## Image registration

For registration of the different channels, cell outlines were extracted from smoothed super-resolved membrane PAINT images using a custom-written macro (Macro M2). MreB and DNA channels were manually translated into these outlines, minimizing the amount of signal outside the cell outlines.

## Cell averaging (CLSM images)

Cell averaging was performed using custom-written Fiji macros. First, cells were segmented in the membrane channel of CLSM images and curated for cells that were only partially attached to the surface, as well as merged cells and debris. ROIs were saved and single cells were extracted using Macro M3 according to the following routine: (i) Rotation according to the angle of a fitted ellipse. (ii) Straightening of the cells based on the centroids of 300 nm segments. (iii) Cell centering based on the center of mass. (iv) Normalization according to cell width and length by image rescaling to a fixed size using Macro M4. (v) Averaging using the Fiji tool 'Z-project'.

## Measurement of the relative nucleoid length

Cell and nucleoid lengths were determined using the custom-written Macro M5. This analysis was performed on rotated, aligned and straightened cells, which were generated using Macro M3. The ratio of nucleoid and cell length finally provides the RNL.

## Determination of the MreB intensity distribution

The relative MreB distribution to cell poles and the cell cylinder was quantified in CLSM average images (created in the previous section) using Macro M6. Cell poles and cylinder were defined using the membrane channel (Fig. S9) and respective ROIs were added to the RoiManager in Fiji. The relative intensity in the respective ROIs was calculated by Eq. 1,

$$I_{rel} = \frac{I_{ROI}}{I_{total}} \times \frac{Area_{total}}{Area_{ROI}} \qquad (1)$$

where $I_{rel}$ represents the relative intensity, $I_{ROI}$ the integrated intensity in the selected ROI (either cell poles or cell cylinder), $I_{total}$ the sum of the integrated intensities of pole and cylinder regions, $Area_{total}$ the sum of the respective areas and $Area_{ROI}$ the area of the selected ROI. Error analysis was performed using error propagation according to Eq. 2,

$$\Delta I_{rel} = \sqrt{\left(\frac{Area_{total}}{Area_{ROI}}\right)^2 \times \left(\left(\frac{1}{I_{total}}\right)^2 \times \Delta I_{ROI}^2 + \left(\frac{I_{ROI}}{I_{total}^2}\right)^2 \times \Delta I_{total}^2\right)} \qquad (2)$$

where $\Delta I_{rel}$ represents the standard error of the mean (SEM) of $I_{rel}$, $\Delta I_{ROI}$ the SEM of $I_{ROI}$ and $\Delta I_{total}$ the SEM of $I_{total}$. $\Delta I_{ROI}$ and $\Delta I_{total}$ were measured in standard deviation images created during the averaging procedure.

## Determination of the RID in SMLM images

Bacterial outlines were smoothed and selected based on the Nile Red channel of multichannel SMLM images (Macro M2). The resulting ROIs were converted into a binary mask and masks of individual *E. coli* cells were eroded iteratively, radially reducing the binary mask in 40 nm steps (cell diameter hence shrinks by 80 nm per step). Resulting areas were transferred to the RoiManager and the integrated intensity in each ROI was measured both in the Nile Red membrane and JF$_{646}$-

Hoechst DNA channel. Erosion and intensity measurements were performed using Macro M7. The relative signal intensity $I_{rel}$ in the removed area was calculated by Eq. 3,

$$I_{rel} = \frac{I_N - I_{N+1}}{I_{total}} \qquad (3)$$

with $I_N$ representing the integrated intensity of a bacterial section $N$, $I_{N+1}$ the integrated intensity of the consecutive section created by erosion of $N$ and $I_{total}$ the integrated intensity within the whole bacterial outline. The relative area was determined by normalizing the area of each ROI created by the erosion procedure (not the removed area) to the total area of the cell outline. Results for each erosion step were determined automatically and saved as text file. Data was finally plotted and visualized using OriginPro 9.1 G (OriginLabs).

## Determination of RID center of mass and FWHM

The centers of mass and FWHM values for all time points and conditions were extracted from RID plots using the integration tool in OriginPro 9.1 G.

## Measurement of intensity plots in population averages

Intensity line plots were generated in population averages in Fiji by averaging the signal for the entire image width (cross-axis plots) or image height (length-axis plot). Standard deviations were extracted similarly from standard deviation images created during the averaging procedure. Intensity traces were further processes in OriginPro 9.1 G.

## Measurement of DNA-membrane distances

DNA-membrane distances were determined from line plots drawn from the cell outside towards the inside. As cutoff-values, we used the maximal value for the membrane signal and max/e (-36.8%). Values were automatically extracted using a custom Python script that can be found in the Github repository.

## Statistical analysis

Descriptive statistics was determined using Origin Pro 9.1 G. For statistical testing of DNA-membrane distances (Fig. 5), we used a two-tailed, unpaired Welch's *t*-test implemented in the SuperPlotsOfData web application[87].

## Circular cross-correlation

Intensity traces of DNA and MreB signal were extracted from the triple-color images based on the cell outlines using Macro M8. In order to measure the signal close to the membrane, cell outlines were eroded by 20 px and signal was measured along the perimeter with a line width of 30 px. Circular cross-correlation was then calculated in Fourier space using *z*-score normalized data. For each condition, the cross-correlations for shift distances of $-2\,\mu m$ to $2\,\mu m$ were averaged and plotted together with the standard error of the mean. The notebook for circular cross-correlation is provided as Supplementary Software and can be found in the Github repository (https://github.com/CKSpahn/Bacterial_image_analysis).

## Randomization of MreB distribution

To randomize MreB distributions, peaks were detected by a Laplacian of Gaussians for each trace and fitted using a GMM. The individual components were randomly distributed along the perimeter. The cross-correlation of the GMM and DNA intensity trace was compared to the cross-correlation of the original data. Only cells with a deviation of >3% and visually proper fits were used for downstream analysis. The circular cross-correlation was calculated between simulated MreB traces and DNA similar to the original data.

## Simulation of images

To validate the circular cross-correlation approach, we simulated 3-color images using the custom-written Fiji Macro M9. We simulated a circle (membrane signal). 8 or 12 spots were positioned equidistantly and slightly towards the circle center to mimic signals of MreB assemblies and DNA. Images were blurred using a Gaussian with a sigma of 4 pixels. To modulate the cross-correlation, MreB and DNA spots were shifted with respect to each other and intensity profiles of simulated images were extracted using Macro M8.

## Training the model to segment DNA in SMLM images

For segmentation of super-resolved DNA in PAINT images, a content-aware image restoration (CARE) model was trained using the ZeroCostDL4Mic platform[88,89]. A deep learning model is beneficial as single-molecule data has a lot of noise, coming from false-positive localizations, but images are also not always homogeneously excited. To generate a model for DNA segmentation, we annotated images of bacteria treated with different antibiotics for different durations. For this, patches of reconstructed images were segmented manually using Fiji and ground-truth masks and image patches were saved separately. The CARE model was then trained on 50 annotated images for 100 epochs with a patch size of $256 \times 256$ px$^2$, 4 patches per image, a batch size of 4, 4-fold data augmentation, 332 steps per epoch, 10% validation data split, and an initial learning rate of 0.0004. The prediction of the model was thresholded in Fiji using Otsu's method and DNA-populated area was measured within the cell outlines provided by membrane PAINT images. The trained model is available within the DeepBacs collection on Zenodo (Table S4).

## Reporting summary

Further information on research design is available in the Nature Portfolio Reporting Summary linked to this article.

## Data availability

Confocal imaging data and multicolor super-resolution images for all conditions are available via Zenodo. The CLSM images generated in this study have been deposited in the Zenodo database under accession 8430052. The data of length- and cross-axis plots have been deposited in the Zenodo database under accession code 14967865. SMLM image data of *E. coli* have been deposited in the Zenodo database under accession code 8430032. SMLM image data of *X. doucetiae* have been deposited in the Zenodo database under accession code 10007398. SMLM image data of *E. coli* RNase E experiments have been deposited in the Zenodo database under accession code 14962042. SMLM nucleoid segmentation data and model have been deposited in the Zenodo database under accession code 8429932. Raw single-molecule localization data will be shared upon request due to its extensive size. Source data are provided with this paper.

## Code availability

Custom-written macros and notebooks (Table S3) are provided as Supplementary Software and in our Github repository (https://github.com/CKSpahn/Bacterial_image_analysis). The current version of this repository has been deposited on Zenodo: https://doi.org/10.5281/zenodo.14968246[90].

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

## Acknowledgements

C.S. and M.H. acknowledge funding by the Deutsche Forschungsgemeinschaft (German Research Foundation; DFG), grants HE 6166/17-1 and SFB 1507. C.S. further acknowledges support by the European Molecular Biology Organization (EMBO) in form of a Scientific Exchange Grant (grant no. 8587) and by the State of Bavaria via the Distinguished Professorship Program 3.1-3122.05-008/014 P000122427. H.B.B. and C.S. acknowledge funding by the Max-Planck Society. R.H. and, E.G.M. acknowledge the support of the Gulbenkian Foundation (Fundação Calouste Gulbenkian), the European Research Council (ERC) under the European Union's Horizon 2020 research and innovation program (grant agreement no. 101001332) and the European Union through the Horizon Europe program (AI4LIFE project with grant agreement 101057970-AI4LIFE, and RT-SuperES project with grant agreement 101099654-RT-SuperES), the European Molecular Biology Organization (EMBO) Installation Grant (EMBO-2020-IG-4734 to R.H.) and Postdoctoral Fellowship (EMBO ALTF 174-2022 to E.G.M.). This work is funded by the European Union. Views and opinions expressed are however those of the author(s) only and do not necessarily reflect those of the European Union. Neither the European Union nor the granting authority can be held responsible for them. E.G.M. acknowledges funding by Fundação para a Ciência e Tecnologia, Portugal (2023.09182.CEECIND/CP2854/CT0004). S.M. was funded by a Biological Sciences Research Council (BBSRC) doctoral studentship (BB/M011186/1). S.H. acknowledges funding from a Wellcome Trust & Royal Society Sir Henry Dale Fellowship [206670/Z/17/Z]. C.S. and R.H. thank Ki Hng (Light microscopy facility at the LMCB, UCL, London) for assistance with the Zeiss Elyra PS1 system. We further kindly thank Zemer Gitai for providing the *E. coli* strain NO34. pmScarlet-i_C1 was a gift from Dorus Gadella (Addgene plasmid # 85044; http://n2t.net/addgene: 85044; RRID:Addgene_85044). pmMaple3-CAM was a gift from Xiaowei Zhuang (Addgene plasmid # 101148; http://n2t.net/addgene:101148; RRID:Addgene_101148). We also thank Agamemnon Carpousis (Toulouse Biotechnology Institute) for providing the RNase E strains, as well as Benoit Lelandais (Pasteur Institute, Paris) for help with the GMM approach.

## Author contributions

C.S. and M.H. conceived the study. C.S. and S.M. acquired microscopy data. C.S., E.G.d.M. wrote code and analyzed the data. M.H., C.S., S.H., R.H., and H.B.B. acquired funding and helped supervise the project. C.S. wrote the article with the input of all authors.

## Funding

## Competing interests

The authors declare no competing interests.

## Additional information

[1]Institute of Physical and Theoretical Chemistry, Goethe University Frankfurt, Frankfurt, Germany. [2]Department of Natural Products in Organismic Interaction, Max Planck Institute for Terrestrial Microbiology, Marburg, Germany. [3]Rudolf Virchow Center for Integrative and Translational Bioimaging, University of Würzburg, Würzburg, Germany. [4]Centre for Bacterial Cell Biology, Newcastle University Biosciences Institute, Faculty of Medical Sciences, Newcastle upon Tyne, UK. [5]Optical cell biology group, Instituto Gulbenkian de Ciência, Oeiras, Portugal. [6]Optical cell biology group, Gulbenkian Institute of Molecular Medicine, Oeiras, Portugal. [7]AI-driven Optical Biology, Instituto de Tecnologia Química e Biológica António Xavier, Universidade Nova de Lisboa, Oeiras, Portugal. [8]UCL-Laboratory for Molecular Cell Biology, University College London, London, UK. [9]Department of Biosciences, Molecular Biotechnology, Goethe University Frankfurt, Frankfurt, Germany. [10]Center for Synthetic Microbiology (SYNMIKRO), Phillips University Marburg, Marburg, Germany. [11]Senckenberg Gesellschaft für Naturforschung, Frankfurt, Germany. [12]Department of Chemistry, Phillips University Marburg, Marburg, Germany. [13]School of Life Sciences, University of Warwick, Gibbet Hill Campus, Coventry, UK. ✉e-mail: Christoph.spahn@uni-wuerzburg.de; heilemann@chemie.uni-frankfurt.de

