## [Transparent peer review file · Nature Communications]

The nucleoid of rapidly growing *Escherichia coli* localizes close to the inner membrane and is organized by transcription, translation and cell geometry

Corresponding Author: Dr Christoph Spahn

Version 0:

Reviewer comments:

Reviewer #1

(Remarks to the Author)

Summary: In this work, Spahn et al imaged the bacterial nucleoid using a combination of multi-color, three dimensional (3D) single-molecule localization superresolution microscopy (SMLM), vertical cell imaging with VerCINI, and confocal scanning laser microscopy. This paper builds upon their 2019 work in which they observed the exclusion of the *E. coli* nucleoid from the center of the cell and the association of the nucleoid with the membrane using STED. In this work the authors show that this membrane association only occurs in rich media and not minimal media. Inhibition of transcription or translation causes the nucleoid to detach from the membrane within minutes, which leads them to suggest that transertion could be a factor in this nucleoid morphology. To explore the role of transertion and cell geometry on nucleoid organization, they use several perturbations, including disrupting MreB filament polymerization, side cell wall synthesis, SecA translocation, in addition to inhibiting transcription and translation. These treatments all had unique effects on nucleoid morphology that were expanded upon in the rest of the study. From these morphological observations, facilitated by their improved resolutions, the authors concluded that (1) the nucleoid adopts a membrane-associated configuration during rich growth conditions; (2) transertion is a main driver of nucleoid positioning and its attachment to the membrane; (3) cell geometry and entropy also play an important role in nucleoid organization and positioning; (4) nucleoid compaction is mainly by transcription. The authors also explored nucleoid organization in the larger species *Xenorhabdus doucetiae*. Overall, the images are stunning; the improved resolutions of nucleoid images across many conditions are impressive. However, the work is largely phenomenological and descriptive. The reviewers have a few major concerns about the authors' conclusions and analyses.

Major Comments:

1. In the superresolution images, the authors observed that DNA is next to the membrane at multiple points. This observation alone does not prove that the chromosomal DNA is membrane associated. Chromosomal DNA can be physically next to the membrane without membrane association, for example, in cells with expanded nucleoids when treating cells with rifampicin or by deleting some of the nucleoid-associated proteins. Treating cells with chloramphenicol caused a major compaction of the nucleoid, which is consistent with the transertion model, but not a proof. To strengthen (or tone down) this conclusion:

a. Can some genetic mutations be used to specifically perturb transertion, other than the global transcription and translation shutdown? The global assault to cell's transcription and translation machineries causes stress responses especially in the long time scales. Can the authors deplete SecA (or a related protein if SecA is essential), or overexpress RNase to degrade mRNAs quickly and see if the treatments have the same effect? Or overexpressing a transertion membrane protein at multiple chromosomal points and subsequently observing the chromosome attachment coupled with smFISH for gene locations?

b. Please quantify the distances between the edge of DNA intensity and the edge of membrane. The improved resolutions may allow the authors to determine the distance more accurately than before, which can then be used as a parameter to examine its variations under different conditions. Will the distance be under 20 nm? 50 nm? And how does it vary along the membrane? Similar characterizations could be conducted for MreB and DNA as well.

2. The correlation between MreB and DNA: the random control for the circular cross-correlation should not be the "reverse"

control, but a randomly scrambled MreB localization along the membrane. How high is the measured cross correlation against this computationally scrambled random colocalization? If the authors express more MreB will they observe higher correlation? What about the conditions where gyrase is inhibited or the elongasome is inhibited? Or MreB is depolymerized? What's the biological significance of this colocalization?

Minor Concerns:

3. While the imaging in this paper is very convincing, the radial distribution of the nucleoid in fast growth has not been directly observed before. These results could be more easily compared to other previous publications if a diffraction limited image of the fixed cells is included in the supplemental figures. A 3D structured illumination microscopy (SIM) image would also be a useful comparison since that is a common method for imaging nucleoid morphology. This would provide more evidence that these exciting results emerge solely from improved resolution.

4. With the image iv in Figure 1A, the void in the middle of the DNA looks slightly larger than the live cells. It is hard to tell if this is completely due to a difference resolution or if the nucleoid associated proteins possibly localize further towards the interior of the cell in these conditions. Some resolution calculations in the supplement to show the larger void in image iv in Figure 1A is plausibly due to resolution would be sufficient to address this detail.

5. Citations should be provided for sodium azide perturbing protein translocation and for nalidixate perturbing DNA replication specifically. Inhibition of DNA gyrase and topoisomerase IV is more accurately described as a global perturbation of supercoiling. Perturbing the chromosome supercoiling state would affect transcription in addition to replication as well as the overall chromosome conformation more generally. Also was there a reason that novobiocin was not used?

6. The images in *X. doucetiae* suggest that the chromosome detachment from the membrane upon transcription or translation is not specific to *E. coli*. However, the two species differ by more than just their dimensions. The authors should compare the chromosome size to cell volume ratio and should note if there are differences in nucleoid associated proteins between the two species. Without the proper comparison/discussion, this data may not be necessarily required for this work.

7. Comments on clarity:

- a. Line 60: "nucleoids are more condensed". The author meant that the nucleoid is more structured in contrast to the amorphous rif-treated cells. "Condensed" often means "compacted" for smaller nucleoid volume. Fast growing cells have large, expanded nucleoid while slow-growing cells have small, compacted nucleoids.
- b. In line 128, MreB should be introduced as a member of the elongasome, since MreB representing elongasomes is assumed in line 133.
- c. Please include "MreB-superfolderGFP sandwich fusion (MreBsw-sfGFP)" in line 131.
- d. In line 151, please clarify if PBP2 is involved in cell division, elongation, or both.
- e. Please confirm that both the second and third panels in Figure 3C are both MP265 treatment conditions.
- f. In the Figure 6 caption, "how" should be removed from "membranes and DNA are shown in red and yellow how."

(Remarks on code availability)

The code seems to be well commented, but I was not sure what input files the pipeline required. The readme file needs to be more descriptive. It should include an explanation of each file and requirements for the input files at minimum.

Reviewer #2

(Remarks to the Author)

In this study, Spahn et al. study the impacts of cell morphology, growth rate, transcription, and translation on the spatial organization of the *E. coli* nucleoid. The work use perturbations to provide evidence for the transertion hypothesis, in which bacterial membrane-protein-encoding genes are localized to the membrane for coupled transcription, translation, and membrane insertion. Transertion has been demonstrated only in limited cases (Kaval et al., 2023; Libby et al., 2012), and has been suggested as the source of rapid re-positioning in the *E. coli* nucleoid with changes in growth rate and antibiotic treatment (Bakshi et al., 2012; Kannaiah et al., 2019; Matsumoto et al., 2015). However, it is unknown whether it is a more generalized organizing principle.

This article uses super-resolution imaging to see changes in the nucleoid and cell morphology with multi-drug treatments that offer insights into how structural and biosynthetic systems work together in determining nucleoid associations with the cell membrane; the manuscript may indeed provide key insights into the relationships between these processes. The mechanisms of bacterial chromosome organization are of high interest to microbiologists and biophysicists, and this is an impressive and state-of-the-art study in microbial imaging. A particular strength of the study is the high-quality imaging used: excellent confocal microscopy, live-cell VerCINI measurements, and 2D and 3D single-molecule imaging. The data (in particular the 3D super-resolution images in Fig 1 and the 2D super-resolution images in Figs 4, 5, and 6), provide convincing evidence that the membrane association of the chromosome can be seen more clearly in 3D and in perturbed conditions, where the membrane shape pulls the chromosome away from the cell center (Fig. 4B). These high-resolution, 3D studies show morphologies that may be unfamiliar to those who are used to 2D projections of the *E. coli* nucleoid and provide new structural data with which to consider questions like the transertion hypothesis. It is also valuable that the authors made their images freely available. The analysis methods provided also integrate easily with freely available software (ImageJ, ZeroCostDL4Mic) which serves the reproducibility of these results. Overall, this is an impressive study that is a valuable contribution to the field.

The methods are detailed and sufficient for the work to be reproduced. Details of the bacterial strains used in the study are presented in the supplementary methods and treatment information is included in the Methods section.

However, the study could have done more to place the observations into the biological context, and to further test their hypotheses. There are also concerns about the authors' considerations of other interpretations of the data that should be corrected before publication, as detailed below.

Major comments

1. Inconsistencies between the data provided and the results expected under the transertion hypothesis. Interpretations of the rifampicin experiments (L106, L212, L220, Figure 3F, Figure 4B, L412) in the article suggest that contraction of the nucleoid within 2-10 minutes of rifampicin treatment are evidence for transertion. However, during fast growth the main genes expressed will be rRNA genes (up to 85% at the fastest growth in LB of ~21 minute doubling times (Paul et al., 2004), although presumably less in the growth rate shown here). These genes are not translated, and therefore would not be related to transertion. In contrast, in M9, a more diverse set of genes is transcribed, more of which would be transertable mRNA genes. The interpretation of the results of this study, in particular Figure 1 and connections to the transertion hypothesis, must consider this discrepancy. Why would there be more membrane-association from transertion in LB than in M9, with presumably fewer membranes being produced in rich media (Figure 1)? Along these lines, can the authors exclude that the absence of transcription (and its RNA products) upon with rifampicin treatment simply changes the level of compaction of the nucleoid without affecting its interaction with the membrane, as in a non-transertion hypothesis? How would the authors prove this? The changes in nucleoid condensation/compaction with rifampicin treatment are clear from the images (Figures 2,3,4,5) but this does not prove that it is due to loss of transertion.
2. L105: The "membrane-bound" state of the nucleoid cannot be proven given the methods used in this study. How can you exclude that the periphery of the nucleoid may be close to the membrane but not attached to it? Can the methods used in the study directly visualize links between the nucleoid and membrane?
3. L215, 264: The phenotypes here seem slightly different than those previously reported for rifampicin and chloramphenicol in rapid growth (e.g. Nonejuie et al., 2013). The DNA here seems less diffuse and cell-filling in the rifampicin condition, and there are fewer ring-like DNA formations in the chloramphenicol condition. What is the difference in conditions? Considering the difference, what would be the relevant regime to examine the effects of blocked transcription?
4. L290: This statement regarding nucleoid positioning is unconvincing. What if the perturbation simply brings the poles closer to the DNA? The cause and effect relationship is not clear.
5. L296: There is no instant "inhibition of protein biosynthesis" with rifampicin treatment. Translation will occur on existing transcripts for much longer.
6. L300, 446: Entropic effects related to the nalidixate phenotype need to be explained.
7. L344: The effect of rifampicin is earlier claimed to be rapid (within 2 minutes). Why, then, is the demonstration of MreB cross-correlation done for 60 minutes of rifampicin treatment? What is the result for 2 minutes of rifampicin treatment? These figures from S23 should be moved to the main text and explained. Even by eye, the correlation does not seem to be lost within 2 minutes. Similarly for the chloramphenicol treatment. By 60 minutes, the nucleoid is homogeneous throughout the cell so if the MreB is heterogeneous, any associations would obviously be lost.
8. L351: The loss of the 2' and 60' timepoints here in Figure 5 makes comparisons with the results in Figure 4 more difficult.
9. L392: There needs to be more depth of discussion on the expected effects of the transertion hypothesis in M9 vs. LB, and how this relates to results.
10. L415: Would be good to see more explanation on why the rifampicin distribution becomes wider at longer treatment times. Why isn't this interpreted as re-establishing links with the membrane? Of course such an interpretation would not line up with the expected mechanism of the rifampicin action - but, equally the condensation of the nucleoid can't directly be linked to transertion.
11. Overall, the current results and discussion do not fully justify the title "Transertion... organize[s] the E. coli nucleoid...". This should be amended.

Minor comments

1. L96: References are more general discussions of super-resolution microscopy; review and provide examples where super-resolution has been used to look at the bacterial chromosome.
2. L191: The range of cell lengths used for averaging should be noted, as this is an indicator of growth rate and phenotype.
3. Figure 3D: The definitions of pole/cylinder and the plots shown are unclear, especially for chloramphenicol (it seems like the opposite of what is shown?).
4. L280: "Complete overlap." S12 shows three cells, with very good overlap of the signals. However, this degree of overlap is not shown quantitatively. There is a need to formalize the degree of overlap for populations of cells. If this is not necessary for the conclusions, the statement of "complete overlap" should be softened.

(Remarks on code availability)

Image data are available on Zenodo, are organized in a logical manner, and can be opened in ImageJ. The analysis pipeline is available on Github and contains sufficient comments; in addition, the Methods detail which macros were used for which steps in the analysis. The macros that we ran with the data provided worked as expected and were easy to use.

Reviewer #3

(Remarks to the Author)

I co-reviewed this manuscript with one of the reviewers who provided the listed reports. This is part of the Nature

Communications initiative to facilitate training in peer review and to provide appropriate recognition for Early Career Researchers who co-review manuscripts.

(Remarks on code availability)

Reviewer #4

(Remarks to the Author)

In their manuscript, Spahn et al. present intriguing insights into the nucleoid organization of *E. coli* and factors that might contribute to this. Using state-of-the-art imaging approaches, they argue in favor of transertion and cell geometry as major driving forces of subcellular nucleoid structure in rich nutrient conditions. For this, they use a variety of chemical perturbations targeting different cellular processes, followed by imaging at regular time intervals using different imaging approaches and subsequent analysis methods. However, while the resulting images are beautiful and the analysis methods appear sound, evidence supporting their major claims is, at best, indirect (see major comments).

Major comments

-The ring-like nucleoid structure reported in rich conditions in Figure 1 is very interesting, but warrants more experimental scrutiny. For example, vercini measurements of cells stained with DAPI could be performed to validate that the observed pattern is not a consequence of bulkier fusion proteins not being able to enter the center of the nucleoid in LB. Combining the nucleoid markers with a diffuse cytosolic marker would also be helpful, as a baseline for cytosolic space. A problematic aspect is that both attachment or detachment of the nucleoid to the (inner) membrane are not directly demonstrated (as these analyses are only based on proximity in microscopy images). Furthermore, the nucleoid organization observed in LB raises the question of what intracellular components are inside this ring-like structure, ribosomes and/or RNA polymerases? Also relevant is how the authors would explain the difference between M9 and LB, less transertion? And if so, would they expect this in more nutrient-poor growth conditions? Potential mechanisms/reasons?

-Treatment of cells with either chloramphenicol or rifampicin has profound effects on cellular physiology, which go beyond simply blocking transertion. As such, the authors conclusion that transertion is a principal organizer of the nucleoid appears (too) strong. While the authors often exert caution in the main text (e.g., "likely transertion" in line 409 and "potential disruption of transertion" in line 467), this is not the case in the title nor the abstract. In general, more orthogonal and direct evidence is needed to definitively implicate transertion in nucleoid organization and the changes observed upon drug treatment. This could be achieved by directly monitoring transertion machinery (and examining whether this co-localizes with nucleoid regions in close proximity to the membrane) or by modifying the extent of transertion inside cells (e.g., by overexpressing transertion machinery, overexpressing specific membrane proteins that are cotranscriptionally translated and inserted into the membrane, or by using an RNaseE variant that no longer localizes to the membrane and has been reported to stabilize inner-membrane protein mRNAs (PMID 27198188)). Without such additional experimental evidence, the arguments presented in favor of transertion playing a role in nucleoid organization do not exceed that of the existing literature.

-Is the apparent association of MreB with chromosomal DNA not simply a consequence of both not localizing to the cell poles and the constriction site? The authors should check cross-correlations within the nucleoid regions to resolve this. An alternative would be to look at newborn/short cells (as all cells now shown in Figure S2 appear to be already septating). The cell shown in Figure 4E as well as its circular cross-correlation does not provide a strong argument for this association either (given the larger peak at a distance of 1 μm). Also, the circular cross-correlations presented in Figure S23 (at $t = 0$ min, so the controls) are not very convincing either, suggesting that the apparent co-localization is either indirect or simply artefactual (from both components localizing to the same cellular region). In addition, why are the cross-correlations not expressed in an interpretable correlation coefficient (see for example Fig. 2 in PMID 5186725)? This would benefit the interpretation of the strength of the correlation. Also helpful would be cross-correlation benchmarks in the form of a cytosolic and DNA-binding protein. Cross-correlations should also be performed with the raw signal and after different types of convolution for the MreB signal. A final question that remains is why this association would exist or be beneficial (especially given that MreB, at least to our knowledge, is not involved in transcription). Perhaps the authors could speculate on this in the Discussion.

-The authors should better highlight and explain the specific contributions/novelty of the presented imaging approaches in the context of existing literature. Currently, it is unclear what the state-of-the-art imaging approaches offer in terms of novel observations or conclusions, as many of the drug effects (on chromosome organization) have already been described in literature (e.g., PMIDs 25250841, 8763959, 31097704, 11489856, 23623305).

-Potential fixation artefacts are mitigated for native nucleoid organization (using live-cell imaging; Figure 1), but not for any of the chemical treatments.

-Other factors that have been proposed to contribute to nucleoid organization, like macromolecular crowding (e.g., 31155353), solvent quality (e.g., PMID 34186018), electrostatic interactions (e.g., PMID 31629479), DNA supercoiling (e.g., PMID 3183003), or the activity of NAPs (e.g., 31767998) are not considered nor discussed (while almost all of them might be altered during the chemical perturbations).

Minor comments

-Vercini measurements: what are we looking at exactly, different cells? How was the imaging plane chosen? Z-stacks and time-lapse videos would be extremely interesting and relevant to add.

-Is Nile Red a proxy for the inner or outer membrane, or both?

-A quantification of to what extent MreB localization depends on the chosen deconvolution approach would be helpful to assess potential variability introduced by the chosen method.

-While many of the figures show averages and standard deviations, the underlying cell numbers are not indicated. Same for the number of replicate experiments.

-Line 274: does the mention of "relative DNA content" refer to the relative DNA amount or relative positioning of the DNA? Can the super-resolution microscopy provide insight into the absolute or relative amount of DNA inside cells?

-Line 441-445: It was previously shown that cell size correlates with nucleoid size, but inversely with nucleocytoplasmic ratio across species, leading to more nucleoid exclusion of ribosomes (PMID 31150626). This appears to directly oppose the hypothesis of the authors.

-Movement of gene loci to the nucleoid periphery has been shown to depend on transcription and translation, but not on transertion (PMID 31719538). Would this not argue against a strong role for transertion in nucleoid organization?

(Remarks on code availability)

Version 1:

Reviewer comments:

Reviewer #1

(Remarks to the Author)

We appreciate that the authors have performed additional experiments with RNaseE overexpression and provided more rigorous quantification of the DNA-membrane distances and of the MreB-DNA distances. The results match their conclusions better now. We find their responses to our concerns generally thorough and satisfactory. One major suggestion is to use "membrane proximity" or "close to membrane" instead of "membrane-associated" in the manuscript to avoid confusions. For example, the title could be revised as: The nucleoid of rapidly growing Escherichia coli localizes close to the inner membrane and is organized by transcription, translation and cell geometry

Minor Comments:

1. Line 177: Thank you for providing a citation explaining the relationship between sodium azide and protein translocation. However, nalidixate should not be described as primarily disrupting DNA replication. It more broadly changes the supercoiling distribution which affects overall chromosome structure and transcription, in addition to replication. Another note is that Nalidixates and Novobiocin both work for Gram negative cells, but with different mechanisms.
2. Line 418: Additional discussion of the RNase E overexpression results would be helpful. Is it possible that actively translating mRNAs are protected from degradation despite RNase E overexpression?

(Remarks on code availability)

The ReadMe provided enough detail to use the code. One additional improvement would be to include an example file of data, but this is not necessary for other people to use the code.

Reviewer #2

(Remarks to the Author)

The authors have addressed all of our main comments and have made valuable additions to the manuscript. In particular, the authors now provide a more thorough discussion of the biophysical properties demonstrated by their experiments, and how they relate to the role of transertion. The greater discussion of the diverging results in M9 and LB is appreciated, as is the additional discussion of rifampicin and chloramphenicol phenotypes at longer treatment times. The amended title now better describes the claims supported by the work.

Some additional comments: the authors should clarify their reasoning in L542-546, in discussing the results of the RNase E expression experiments. Currently, the paragraph claims that transertion-mediated positioning is possible, and that RNase E localization and abundance does not significantly contribute to nucleoid organization, but the biological link between these statements should be made clearer in this discussion. The responses to comments 2 and 10 also need to be introduced in the ms.

(Remarks on code availability)

Reviewer #3

(Remarks to the Author)

(Remarks on code availability)

Reviewer #4

(Remarks to the Author)

I commend the author's extensive revisions and clarifications. However, I think that several of my main points still stand:

There is no direct evidence that implicates transertion. While the RNaseE experiments are interesting, I was hoping to see a larger effect. For now, these do not add much, besides, as the authors indicate in the study, that "degradosome activity, at least under the condition tested, does not significant contribute to nucleoid organization" (so not much transertion-related insights). Early timepoints of drug treatments can also not be seen as targeted disruptions of transertion. For now, transertion thus remains a hypothesis, and statements regarding it should be amended throughout the text. For example, subtitles such as "Super-resolution microscopy reveals complete nucleoid detachment from the inner membrane upon inhibition of protein biosynthesis" or statements in the discussion like "indicates that active transertion is required to maintain this expanded, membrane-localized state." are somewhat misleading.

Similarly, other factors involved in nucleoid organization were not directly tested, so their suggested non-involvement also remains speculative.

Related to specific contributions/novelty: although I agree with the authors that SMLM increases contrast and resolution, I still struggle to find new biological insights or knowledge that have come from using this approach, and that were previously not accessible from literature (were standard microscopy methods were mostly used).

With regard to the apparent association of MreB with chromosomal DNA, many of my points were not addressed in the rebuttal:

-In Figure 4G, the circular cross-correlation for shorter cells displays a smaller peak than that observed in larger cells, arguing in favor of my previous argument that the apparent co-localization simply stems from both components localizing to the same cellular region (i.e., not the poles nor the division septum).

-Related, in Figure S25 (first column), there is a negative correlation between the cytosolic control and DNA, suggesting exclusion of a free cytosolic protein by the nucleoid. Is this correct, artefactual, or how do the authors interpret this? (given that exclusion of a small, free cytosolic protein by the nucleoid would be unexpected)

-I am still not sure how I should interpret the y-axis in terms of correlation strength. Is a value of 50 (circular x-corr [A.U.]) a lot? Why are these not expressed as interpretable correlation coefficients?

-Why would this association exist? Also, many more membrane protein assemblies exist, why did the author focus on this one?

(Remarks on code availability)

Point-to-point response to the reviewers

We thank all Reviewers for their valuable feedback on our manuscript. Inspired by this, we performed additional experiments and analyses that are included in this revision, and which in our opinion greatly improved the quality of our manuscript. A detailed response to all comments is appended below.

REVIEWER COMMENTS

Reviewer #1 (Remarks to the Author):

Summary: In this work, Spahn et al imaged the bacterial nucleoid using a combination of multi-color, three dimensional (3D) single-molecule localization superresolution microscopy (SMLM), vertical cell imaging with VerCINI, and confocal scanning laser microscopy. This paper builds upon their 2019 work in which they observed the exclusion of the E. coli nucleoid from the center of the cell and the association of the nucleoid with the membrane using STED. In this work the authors show that this membrane association only occurs in rich media and not minimal media. Inhibition of transcription or translation causes the nucleoid to detach from the membrane within minutes, which leads them to suggest that transcription could be a factor in this nucleoid morphology. To explore the role of transcription and cell geometry on nucleoid organization, they use several perturbations, including disrupting MreB filament polymerization, side cell wall synthesis, SecA translocation, in addition to inhibiting transcription and translation. These treatments all had unique effects on nucleoid morphology that were expanded upon in the rest of the study. From these morphological observations, facilitated by their improved resolutions, the authors concluded that (1) the nucleoid adopts a membrane-associated configuration during rich growth conditions; (2) transcription is a main driver of nucleoid positioning and its attachment to the membrane; (3) cell geometry and entropy also play an important role in nucleoid organization and positioning; (4) nucleoid compaction is mainly by transcription. The authors also explored nucleoid organization in the larger species *Xenorhabdus doucetiae*. Overall, the images are stunning; the improved resolutions of nucleoid images across many conditions are impressive. However, the work is largely phenomenological and descriptive. The reviewers have a few major concerns about the authors' conclusions and analyses.

Major Comments:

1. In the superresolution images, the authors observed that DNA is next to the membrane at multiple points. This observation alone does not prove that the chromosomal DNA is membrane associated. Chromosomal DNA can be physically next to the membrane without membrane association, for example, in cells with expanded nucleoids when treating cells with rifampicin or by deleting some of the nucleoid-associated proteins. Treating cells with chloramphenicol caused a major compaction of the nucleoid, which is consistent with the transcription model, but not a proof. To strengthen (or tone down) this conclusion:

Response: We thank the Reviewer for this important feedback. We agree that mere proximity is not a proof of interaction, and that DNA can be close to the membrane independent of interactions. Random membrane localization of DNA can and will occur during decondensation of the nucleoid. Especially at later stages of rifampicin treatment, chromosomal DNA fills the entire cell volume, leading to this random proximity. However, the distribution strongly differs from the observed

proximity in untreated cells, where the majority of chromosomal DNA is positioned close to the membrane and absent in a large fraction of the cell cytosol. This positioning, as well as the effects of antibiotics at this resolution, to our knowledge, was not reported before.

a. Can some genetic mutations be used to specifically perturb transcription, other than the global transcription and translation shutdown? The global assault to cell's transcription and translation machineries causes stress responses especially in the long time scales. Can the authors deplete SecA (or a related protein if SecA is essential), or overexpress RNase E to degrade mRNAs quickly and see if the treatments have the same effect? Or overexpressing a transcription membrane protein at multiple chromosomal points and subsequently observing the chromosome attachment coupled with smFISH for gene locations?

Response: We thank the Reviewer for these valuable suggestions. As the Reviewer correctly points out, SecA is indeed essential, and so are other proteins that might be part of the transcription complex (e.g. RNA polymerase, NusG, YidC); their depletion will have a strong effect on intracellular organization. Overexpression of RNase E is an excellent idea, which we followed by conducting additional experiments. To conduct these experiments, Agamemnon J. Carpousis kindly provided RNase E mutant and expression strains (see <https://doi.org/10.1371/journal.pgen.1004961>). The mutant strain lacking the membrane-targeting sequence (ΔA strain) showed a slightly increased membrane-DNA distance that we attribute to the slower growth of this phenotype (~ 39 min vs 29 min doubling time). The cell width is still mostly populated by the nucleoid, even in the ΔA mutant strain. We also correlated membrane-DNA distance with the RNase E expression level. For this we used a strain that harbors the *rne* gene on a low-copy plasmid under control of the inducible lacI^{PO} promoter. Growing this strain in the presence of varying inducer levels resulted in a 5-fold difference in RNase E expression. We did not find a correlation between RNase E expression level and membrane-DNA distance, as now shown in Figure 6.

The proposed correlative smFISH experiment would be an exciting way to provide direct evidence for transcription. This, however, requires a significant amount of method development that we were not able to provide within the time frame of this revision. While we feel that this experiment is beyond the scope of this work, it will be part of future research.

b. Please quantify the distances between the edge of DNA intensity and the edge of membrane. The improved resolutions may allow the authors to determine the distance more accurately than before, which can then be used as a parameter to examine its variations under different conditions. Will the distance be under 20 nm? 50 nm? And how does it vary along the membrane? Similar characterizations could be conducted for MreB and DNA as well.

Response: We thank the Reviewer for this excellent suggestion. We quantified the membrane-DNA distance using an automated analysis routine. In untreated cells, we found a distance of 125 ± 12 nm. We next analyzed the changes in this distance for the different drug treatments, and added these experiments and analyses as an additional panel to Figure 4 (**Fig. 4E**).

2. The correlation between MreB and DNA: the random control for the circular cross-correlation should not be the "reverse" control, but a randomly scrambled MreB localization along the membrane. How high is the measured cross correlation against this computationally scrambled random colocalization? If the authors express more MreB will they observe higher correlation? What about the conditions where gyrase is inhibited or the elongasome is inhibited? Or MreB is depolymerized? What's the biological significance of this colocalization?

Response: We thank the Reviewer suggesting an alternative approach for the random control. We followed this suggestion and employed a Gaussian Mixture Model fitting, which identifies and fits the MreB signals and redistributes these signals randomly along the cell perimeter (see S24). This control also results in a loss of correlation between the nucleoid and elongasomes, as shown by the circular cross-correlation plots (**Fig. 4**), and strengthens our findings.

It is not evident that the observed proximity between MreB and the nucleoid is mediated by MreB. We rather interpret the data such that the entire elongasome as a multi-protein complex contributes to the observed spatial arrangement. This is supported by early timepoints of MP265 treatments, where MreB mostly depolymerizes, but no significant changes of nucleoid structure were observed. We rephrased several sections in the manuscript to strengthen this point and referred to MreB as a representative protein for the elongation machinery.

Minor Concerns:

3. While the imaging in this paper is very convincing, the radial distribution of the nucleoid in fast growth has not been directly observed before. These results could be more easily compared to other previous publications if a diffraction limited image of the fixed cells is included in the supplemental figures. A 3D structured illumination microscopy (SIM) image would also be a useful comparison since that is a common method for imaging nucleoid morphology. This would provide more evidence that these exciting results emerge solely from improved resolution.

Response: We thank the Reviewer for this suggesting this important control. In order to achieve the best comparability to the data presented in the figures, we calculated the standard deviation image from the single-molecule PAINT movies. This provides, to a very good approximation, a diffraction-limited image. This data was added to **Figure S13**.

As an additional control, we refer to 3D SIM measurements of fixed and DAPI-stained nucleoids that were recorded in our previous work (see <https://doi.org/10.1038/ncomms13711>, Fig S9), a reference we omitted in the original revision. This control measurement also shows a lower DNA concentration at the radial cell center.

4. With the image iv in Figure 1A, the void in the middle of the DNA looks slightly larger than the live cells. It is hard to tell if this is completely due to a difference resolution or if the nucleoid associated proteins possibly localize further towards the interior of the cell in these conditions. Some resolution calculations in the supplement to show the larger void in image iv in Figure 1A is plausibly due to resolution would be sufficient to address this detail.

Response: We thank the reviewer for sharing this observation. We attribute this to the different resolution that both techniques provide, together with a lower background signal in the single-molecule super-resolution data as compared to the live cell data, which leads to an additional blur of in the images. To demonstrate this effect qualitatively, we generated a version of the image in Figure 1Aiv that was computationally adjusted in resolution to match the live cell data, here attached as Figure R1. Resolution adjustment leads to a smaller void, as seen in the live-cell data.

Figure R1: Side view of one sister chromosome of the bacterium shown in Figure 1Aiv. Resolution is digitally reduced by increasing the particle size(Full width at half maximum, FWHM) in density plots. Strong blur results in a smaller void as observed during live-cell imaging.

5. Citations should be provided for sodium azide perturbing protein translocation and for nalidixate perturbing DNA replication specifically. Inhibition of DNA gyrase and topoisomerase IV is more accurately described as a global perturbation of supercoiling. Perturbing the chromosome supercoiling state would affect transcription in addition to replication as well as the overall chromosome conformation more generally. Also was there a reason that novobiocin was not used?

Response: We thank the Reviewer for this comment. We used nalidixate, as it is described as a gyrase inhibitor in Gram-negative cells, whereas novobiocin is described as a specific inhibitor in Gram-positive cells in the literature. We added citations for the effect of the two compounds to the manuscript.

6. The images in *X. doucetiae* suggest that the chromosome detachment from the membrane upon transcription or translation is not specific to *E. coli*. However, the two species differ by more than just their dimensions. The authors should compare the chromosome size to cell volume ratio and should note if there are differences in nucleoid associated proteins between the two species. Without the proper comparison/discussion, this data may not be necessarily required for this work.

Response: We agree that *E. coli* and *X. doucetiae* differ in many aspects. While their genome has approximately the same size, a lot of genetic material is dedicated to the production of natural products or secretion machineries. This is required for their growth in insect prey as well as symbiosis with their host nematode. We found it thus very interesting that the membrane-proximal positioning and response to rifampicin are conserved, despite the different lifestyles.

We now added a brief discussion as Supplementary Note 3. For the major nucleoid associated proteins (except StpA), we found homologues in *X. doucetiae*. We also quantified the nucleoid-to-cell area ratio (relative nucleoid size), which is larger in *X. doucetiae* both in unperturbed and rifampicin-treated cells (Fig. S28).

7. Comments on clarity:

a. Line 60: “nucleoids are more condensed”. The author meant that the nucleoid is more structured in contrast to the amorphous rif-treated cells. “Condensed” often means “compacted” for smaller nucleoid volume. Fast growing cells have large, expanded nucleoid while slow-growing cells have small, compacted nucleoids.

Response: The Reviewer is fully correct. We removed the term compacted, as it was misleading.

- b. In line 128, MreB should be introduced as a member of the elongasome, since MreB representing elongasomes is assumed in line 133.
- c. Please include “MreB-superfolderGFP sandwich fusion (MreBsw-sfGFP)” in line 131.
- d. In line 151, please clarify if PBP2 is involved in cell division, elongation, or both.
- e. Please confirm that both the second and third panels in Figure 3C are both MP265 treatment conditions.
- f. In the Figure 6 caption, “how” should be removed from “membranes and DNA are shown in red and yellow how.”

Response: We thank the Reviewer for pointing this out, and made according corrections in the revised manuscript.

Reviewer #1 (Remarks on code availability):

The code seems to be well commented, but I was not sure what input files the pipeline required. The readme file needs to be more descriptive. It should include an explanation of each file and requirements for the input files at minimum.

Response: We thank the Reviewer for checking the code we provided. We added a table to the repository's readme file, in which we specify the input and output files and provide additional information for each macro.

Reviewer #2 (Remarks to the Author):

In this study, Spahn et al. study the impacts of cell morphology, growth rate, transcription, and translation on the spatial organization of the *E. coli* nucleoid. The work uses perturbations to provide evidence for the transertion hypothesis, in which bacterial membrane-protein-encoding genes are localized to the membrane for coupled transcription, translation, and membrane insertion. Transertion has been demonstrated only in limited cases (Kaval et al., 2023; Libby et al., 2012), and has been suggested as the source of rapid re-positioning in the *E. coli* nucleoid with changes in growth rate and antibiotic treatment (Bakshi et al., 2012; Kannaiah et al., 2019; Matsumoto et al., 2015). However, it is unknown whether it is a more generalized organizing principle. This article uses super-resolution imaging to see changes in the nucleoid and cell morphology with multi-drug treatments that offer insights into how structural and biosynthetic systems work together in determining nucleoid associations with the cell membrane; the manuscript may indeed provide key insights into the relationships between these processes. The mechanisms of bacterial chromosome organization are of high interest to microbiologists and biophysicists, and this is an impressive and state-of-the-art study in microbial imaging. A particular strength of the study is the high-quality imaging used: excellent confocal microscopy, live-cell VerCINI measurements, and 2D and 3D single-molecule imaging. The data (in particular the 3D super-resolution images in Fig 1 and the 2D super-resolution images in Figs 4, 5, and 6), provide convincing evidence that the membrane association of the chromosome can be seen more clearly in 3D and in perturbed conditions, where the membrane shape pulls the chromosome away from the cell center (Fig. 4B). These high-resolution, 3D studies show morphologies that may be unfamiliar to those who are used to 2D projections of the *E. coli* nucleoid and provide new structural data with which to consider questions

like the transertion hypothesis. It is also valuable that the authors made their images freely available. The analysis methods provided also integrate easily with freely available software (ImageJ, ZeroCostDL4Mic) which serves the reproducibility of these results. Overall, this is an impressive study that is a valuable contribution to the field.

The methods are detailed and sufficient for the work to be reproduced. Details of the bacterial strains used in the study are presented in the supplementary methods and treatment information is included in the Methods section.

However, the study could have done more to place the observations into the biological context, and to further test their hypotheses. There are also concerns about the authors' considerations of other interpretations of the data that should be corrected before publication, as detailed below.

Response: We thank the Reviewer for the positive evaluation of our work. We agree that the insights provided by the increased resolution are exciting and also understand the concerns regarding the biological interpretations. We addressed all points raised by the Reviewer, and made according changes to the manuscript.

Major comments

1. Inconsistencies between the data provided and the results expected under the transertion hypothesis. Interpretations of the rifampicin experiments (L106, L212, L220, Figure 3F, Figure 4B, L412) in the article suggest that contraction of the nucleoid within 2-10 minutes of rifampicin treatment are evidence for transertion. However, during fast growth the main genes expressed will be rRNA genes (up to 85% at the fastest growth in LB of ~21 minute doubling times (Paul et al., 2004), although presumably less in the growth rate shown here). These genes are not translated, and therefore would not be related to transertion. In contrast, in M9, a more diverse set of genes is transcribed, more of which would be transertable mRNA genes. The interpretation of the results of this study, in particular Figure 1 and connections to the transertion hypothesis, must consider this discrepancy. Why would there be more membrane-association from transertion in LB than in M9, with presumably fewer membranes being produced in rich media (Figure 1)?

Along these lines, can the authors exclude that the absence of transcription (and its RNA products) upon with rifampicin treatment simply changes the level of compaction of the nucleoid without affecting its interaction with the membrane, as in a non-transertion hypothesis? How would the authors prove this? The changes in nucleoid condensation/compaction with rifampicin treatment are clear from the images (Figures 2,3,4,5) but this does not prove that it is due to loss of transertion.

Response: We thank the Reviewer for pointing out the differences in transcription dynamics between different media / growth conditions. We agree that this has been not clearly enough discussed, and we now include more information in the revised manuscript. Regarding the difference between M9 and LB, there are several explanations. Transcripts were found to be more stable during slow growth. Translation might thus occur on completed transcripts that are not associated with the DNA. Also, the fraction of inner membrane proteins of the total proteome does not change with growth rate. Fast growth thus requires enhanced inner membrane expression dynamics, which might lead to a stronger contribution of transertion. Regarding a non-transertion effect of transcription inhibition: Rifampicin- and chloramphenicol-treatments show similar phenotypes during early time points (2-10 min). If the absence of transcription and transcripts would increase nucleoid compaction, we would expect a different phenotype during chloramphenicol treatment.

2. L105: The “membrane-bound” state of the nucleoid cannot be proven given the methods used in this study. How can you exclude that the periphery of the nucleoid may be close to the membrane but not attached to it? Can the methods used in the study directly visualize links between the nucleoid and membrane?

Response: We don't have a direct proof of the membrane-bound state of the nucleoid. We interpret the fast response during inhibition of transcription or translation as detachment from the membrane, as other mechanisms, to us, do not explain the observed responses. Such mechanisms could be changes in solvent quality by RNA degradation or changes in polymer rigidity, both able to cause changes on the minute timescale. While RNA degradation occurs rapidly during rifampicin treatment, it does not explain the rapid nucleoid contraction during chloramphenicol treatment, where transcription is largely not affected. As we observe similar short-term effects for rifampicin and chloramphenicol treatments, we consider the mentioned mechanisms as less likely. Unfortunately, the presented methods cannot be used to visualize a direct link between the membrane and nucleoid. Including imaging of genomic loci and/or mRNA into the existing workflow could provide more direct evidence and is part of the future route of the lab.

3. L215, 264: The phenotypes here seem slightly different than those previously reported for rifampicin and chloramphenicol in rapid growth (e.g. Nonejuie et al., 2013). The DNA here seems less diffuse and cell-filling in the rifampicin condition, and there are fewer ring-like DNA formations in the chloramphenicol condition. What is the difference in conditions? Considering the difference, what would be the relevant regime to examine the effects of blocked transcription?

Response: We thank the Reviewer for this important comment. The assay performed by Nonejuie et al. differs from our assay by antibiotic concentrations (0.025 µg/ml rifampicin and 10 µg/ml for chloramphenicol) and treatment times (2h). We also observe nucleoid expansion at longer incubation times of rifampicin (e.g. 1h time point in Figure 4), and it might become even more condensed with longer incubation time. Regarding the relevant regime, we think that the early time points of treatment (2-10 min) are more relevant when studying the direct effects of transcription on nucleoid structure. Longer exposure times can cause diverse downstream effects which are induced by the dilution of proteins, which can indirectly affect cell and nucleoid morphology.

4. L290: This statement regarding nucleoid positioning is unconvincing. What if the perturbation simply brings the poles closer to the DNA? The cause and effect relationship is not clear.

Response: Nucleoids typically populate the quarter positions during rapid growth, both in large and small cells, and avoid the pole regions. The rod-to-sphere transition caused by MP265 or Mecillinam changes this organization with the nucleoid also occupying the previously free space at the poles. This effect was also observed by Japaridze and colleagues (<https://doi.org/10.1038/s41467-020-16946-7>), which we now cite. We agree that the ultimate cause of this reorganization is unclear, but confinement and membrane curvature are likely candidates.

5. L296: There is no instant “inhibition of protein biosynthesis” with rifampicin treatment. Translation will occur on existing transcripts for much longer.

Response: We agree with the Reviewer and changed the section according. We were referring to protein expression from DNA-associated mRNA.

6. L300, 446: Entropic effects related to the nalidixate phenotype need to be explained.

Response: We apologize for the negligence, and now explain entropic effects in this context in more detail in the revised manuscript.

7. L344: The effect of rifampicin is earlier claimed to be rapid (within 2 minutes). Why, then, is the demonstration of MreB cross-correlation done for 60 minutes of rifampicin treatment? What is the result for 2 minutes of rifampicin treatment? These figures from S23 should be moved to the main text and explained. Even by eye, the correlation does not seem to be lost within 2 minutes. Similarly for the chloramphenicol treatment. By 60 minutes, the nucleoid is homogeneous throughout the cell so if the MreB is heterogeneous, any associations would obviously be lost.

Response: We thank the Reviewer for this comment and moved more all panels of rifampicin and chloramphenicol treatments to the main text. As we described in the manuscript, the association is lost gradually, which can be observed both in the in cross-correlation plots and the super-resolution images.

8. L351: The loss of the 2' and 60' timepoints here in Figure 5 makes comparisons with the results in Figure 4 more difficult.

Response: We chose the timepoints with the strongest reorganization for this combinatorial experiment, based on the individual antibiotic treatments.

9. L392: There needs to be more depth of discussion on the expected effects of the transertion hypothesis in M9 vs. LB, and how this relates to results.

Response: We thank the reviewer for this important comment and want to refer the comment of Reviewer #1. The difference between M9 and LB are now discussed more in detail.

10. L415: Would be good to see more explanation on why the rifampicin distribution becomes wider at longer treatment times. Why isn't this interpreted as re-establishing links with the membrane? Of course such an interpretation would not line up with the expected mechanism of the rifampicin action - but, equally the condensation of the nucleoid can't directly be linked to transertion.

Response: Nucleoid expansion at long rifampicin treatments was observed before and sometimes considered as the main phenotype, especially in morphological fingerprinting and antimicrobial susceptibility tests (e.g. <https://doi.org/10.1073/pnas.1311066110>). As transcription inhibition affects the levels of RNAs and proteins systematically, many downstream effects occur that can alter nucleoid morphology. Re-establishment of membrane links might be possible, but is hard to differentiate from random proximity of the entirely decondensed nucleoid. However, Weber and colleagues found that the mobility of genetic loci increases particularly during rifampicin treatment (<https://doi.org/10.1103/PhysRevLett.104.238102>). We thus conclude that the nucleoid at longer rifampicin exposure times is in a decondensed and highly mobile state, in which the proximity to the membrane is of random nature. We now discuss this in the manuscript.

11. Overall, the current results and discussion do not fully justify the title "Transertion... organize[s] the E. coli nucleoid...". This should be amended.

Response: We agree that our experiments do not provide direct evidence for a membrane-bound state. We thus changed the title of the manuscript, referring to the expanded state of the nucleoid that is disrupted by transcription and translation inhibition. We still think that transertion is the process that explains our observations best. We discuss this now in more detail.

Minor comments

1. L96: References are more general discussions of super-resolution microscopy; review and provide examples where super-resolution has been used to look at the bacterial chromosome.

Response: We followed the suggestion of the Reviewer and now include references for 3D-SIM measurements in *E. coli* and *B. subtilis*, as well as some specific studies on cell-wall synthesis.

2. L191: The range of cell lengths used for averaging should be noted, as this is an indicator of growth rate and phenotype.

Response: We agree that this is useful information that was missing. In the revised manuscript, we added information on cell count and length for all conditions in Table S3.

Individual cells of all lengths were used in the averaging procedure. As we have shown in Fig. S4, cell normalization provides average images that only differ slightly in sister chromosome sub-structuring.

3. Figure 3D: The definitions of pole/cylinder and the plots shown are unclear, especially for chloramphenicol (it seems like the opposite of what is shown?).

Response: We verified the assignment in Figure 3D again. We can confirm that for all conditions, cell poles and cylinder were determined the same way, and that all assignments are correct. We apologize if we might have misunderstood this comment.

4. L280: "Complete overlap." S12 shows three cells, with very good overlap of the signals. However, this degree of overlap is not shown quantitatively. There is a need to formalize the degree of overlap for populations of cells. If this is not necessary for the conclusions, the statement of "complete overlap" should be softened.

Response: We agree with the Reviewer that the statement of a "complete overlap" requires quantification. We followed the suggestion by the Reviewer and softened the statement.

Reviewer #2 (Remarks on code availability):

Image data are available on Zenodo, are organized in a logical manner, and can be opened in ImageJ. The analysis pipeline is available on Github and contains sufficient comments; in addition, the Methods detail which macros were used for which steps in the analysis. The macros that we ran with the data provided worked as expected and were easy to use.

Response: We thank the Reviewer for taking the time to review the data availability as well as the functionality of the macros.

Reviewer #3 (Remarks to the Author):

Response: We thank the Reviewer 3 for co-reviewing our manuscript.

Reviewer #4 (Remarks to the Author):

In their manuscript, Spahn et al. present intriguing insights into the nucleoid organization of *E. coli* and factors that might contribute to this. Using state-of-the-art imaging approaches, they argue in favor of transertion and cell geometry as major driving forces of subcellular nucleoid structure in rich nutrient conditions. For this, they use a variety of chemical perturbations targeting different cellular processes, followed by imaging at regular time intervals using different imaging approaches and subsequent analysis methods. However, while the resulting images are beautiful and the analysis methods appear sound, evidence supporting their major claims is, at best, indirect (see major comments).

Response: We thank the Reviewer for taking the time to rigorously evaluate our manuscript and appreciating the quality of our images.

We acknowledge that the evidence we provide is of indirect nature, however at the same time, we believe that our data provides a new angle of experimental insights that at the same time convincingly support the mechanism of transertion. So far, transertion was more directly visualized in the context of secretion machineries (see <https://doi.org/10.1038/s41467-023-36762-z>). A direct proof of transertion at the nucleoid level is still difficult to realize, which is one reason that this hypothesis is still debated in the community. In this work, nano-scale imaging and functional manipulation of some of the key cellular players together with novel tools of bacteria-tailored quantitative image analysis present a new view on nucleoid organization. This was further strengthened by additional experiments that we conducted in the frame of this revision, following the suggestions by the Reviewer.

Furthermore, the technological developments can be transferred to other studies that aim to quantify cellular processes in bacterial cells with nano-scale spatial resolution.

Major comments

-The ring-like nucleoid structure reported in rich conditions in Figure 1 is very interesting, but warrants more experimental scrutiny. For example, vercini measurements of cells stained with DAPI could be performed to validate that the observed pattern is not a consequence of bulkier fusion proteins not being able to enter the center of the nucleoid in LB. Combining the nucleoid markers with a diffuse cytosolic marker would also be helpful, as a baseline for cytosolic space. A problematic aspect is that both attachment or detachment of the nucleoid to the (inner) membrane are not directly demonstrated (as these analyses are only based on proximity in microscopy images). Furthermore, the nucleoid organization observed in LB raises the question of what intracellular components are inside this ring-like structure, ribosomes and/or RNA polymerases? Also relevant is how the authors would explain the difference between M9 and LB, less transertion? And if so, would they expect this in more nutrient-poor growth conditions? Potential mechanisms/reasons?

Response: We thank the Reviewer for these helpful suggestions. In the frame of this revision, we conducted additional experiments and imaged fixed, DAPI-stained cells, which showed a similar phenotype. The data is added as Supplementary Figure 2. We did not perform live-cell imaging, since DAPI is known to affect nucleoid morphology and cell viability (see <https://doi.org/10.3791%2F56497>).

Regarding the content of the DNA-free space, we assume that it is filled with cytosolic components such as proteins, (see Supplementary Figure 26; cytosolic PAmCherry1), RNAs, ribosomes and others. Several studies reported an exclusive staining pattern of ribosomes and the nucleoid, e.g. work by the Jacobs-Wagner or Kapanidis labs (e.g. <https://doi.org/10.1016/j.cell.2021.05.037> and <https://doi.org/10.1101/2024.06.18.24309111>). We thank the Reviewer for pointing us at this important point, and we added this information to the discussion of the revised manuscript. We also discuss differences between LB and M9 (for this, we kindly refer to the response to a similar comment by Reviewer #2).

-Treatment of cells with either chloramphenicol or rifampicin has profound effects on cellular physiology, which go beyond simply blocking transertion. As such, the authors conclusion that transertion is a principal organizer of the nucleoid appears (too) strong. While the authors often exert caution in the main text (e.g., “likely transertion” in line 409 and “potential disruption of transertion” in line 467), this is not the case in the title nor the abstract. In general, more orthogonal and direct evidence is needed to definitively implicate transertion in nucleoid organization and the changes observed upon drug treatment. This could be achieved by directly monitoring transertion machinery (and examining whether this co-localizes with nucleoid regions in close proximity to the membrane) or by modifying the extent of transertion inside cells (e.g., by overpressing transertion machinery, overexpressing specific membrane proteins that are cotranscriptionally translated and inserted into the membrane, or by using an RNaseE variant that no longer localizes to the membrane and has been reported to stabilize inner-membrane protein mRNAs (PMID 27198188)). Without such additional experimental evidence, the arguments presented in favor of transertion playing a role in nucleoid organization do not exceed that of the existing literature.

Response: We thank the Reviewer for these helpful suggestions. At the same time, we agree that our conclusions were mainly drawn from the analysis of the super-resolved images. We followed the suggestion of the Reviewer and conducted new experiments using the cytosolic RNaseE mutant. In these experiments, we did not observe major changes in nucleoid architecture; minor changes, such as an increase in DNA-membrane distance, are likely the result of the slower growth phenotype of the mutant. As such, these experiments strengthen our findings. We thank the Reviewer for suggesting these experiments, which are indeed an elegant functional assay; this new data is now included in Figure 6.

In addition, we attribute the gradual detachment observed during drug treatment are strong indicators for an indirect coupling to the membrane, yet we agree that biochemical or genetic evidence would strengthen our hypothesis. The visualization of the transertion machinery together with the nucleoid is a major goal for our future research, yet is so far not possible with existing imaging technologies for bacterial cells (e.g., where multiplexed super-resolution imaging is much less established and has additional technical limitations in bacteria, as compared to eukaryotic cells).

-Is the apparent association of MreB with chromosomal DNA not simply a consequence of both not localizing to the cell poles and the constriction site? The authors should check cross-correlations within the nucleoid regions to resolve this. An alternative would be to look at newborn/short cells (as all cells now shown in Figure S2 appear to be already septating). The cell shown in Figure 4E as well

as its circular cross-correlation does not provide a strong argument for this association either (given the larger peak at a distance of 1 μm). Also, the circular cross-correlations presented in Figure S23 (at $t = 0$ min, so the controls) are not very convincing either, suggesting that the apparent co-localization is either indirect or simply artefactual (from both components localizing to the same cellular region). In addition, why are the cross-correlations not expressed in an interpretable correlation coefficient (see for example Fig. 2 in PMID 5186725)? This would benefit the interpretation of the strength of the correlation. Also helpful would be cross-correlation benchmarks in the form of a cytosolic and DNA-binding protein. Cross-correlations should also be performed with the raw signal and after different types of convolution for the MreB signal. A final question that remains is why this association would exist or be beneficial (especially given that MreB, at least to our knowledge, is not involved in transcription). Perhaps the authors could speculate on this in the Discussion.

Response: We thank the Reviewer for suggesting these control experiments.

We followed this suggestion and measured the cross-correlation in cells of different sizes using the pooled dataset of untreated cells (270 cells). We split the data into 3 subsets based on their perimeter, which is an indicator of cell length. We did observe the center peak for all length intervals (Fig. 4G). We chose the circular cross-correlation as it preserves the spatial context. For example, there are cases where elongasomes and nucleoids are attaching but do not show maximal overlap. Such cases are visualized in the cross-correlation plot and would be less prominent when providing a single value. The idea of using benchmarks is excellent, so we performed the circular cross-correlation analysis on control data from one of our previous publications (Spahn et al. 2018, Sci. Rep.). These controls show a correlation for nucleoid-associated RpoC-PAmCherry and an anti-correlation for cytosolic PAmCherry. This, to us, confirms that the cross-correlation approach provides a qualitative measure for spatial proximity, even in the absence of strict colocalization.

-The authors should better highlight and explain the specific contributions/novelty of the presented imaging approaches in the context of existing literature. Currently, it is unclear what the state-of-the-art imaging approaches offer in terms of novel observations or conclusions, as many of the drug effects (on chromosome organization) have already been described in literature (e.g., PMIDs 25250841, 8763959, 31097704, 11489856, 23623305).

Response: We thank the reviewer for this suggestion. We agree that the main phenotypes of the drug treatments were already described, but our high-quality and high-resolution images allow more accurate measurements and reveal details that are not accessible with conventional methods. In particular, the precise mapping of protein and DNA localization at various time points of drug treatment benefits from the increased contrast and resolution provided by SMLM. We added this to the discussion.

-Potential fixation artefacts are mitigated for native nucleoid organization (using live-cell imaging; Figure 1), but not for any of the chemical treatments.

Response: We thank the Reviewer for pointing this out. In order to assess fixation artefacts, we conducted super-resolution and live-cell microscopy experiments for untreated cells. We found a good structural agreement, which suggests that the chemical fixation strategy we used does not lead to an observable structural change. This fixation strategy was *bona fide* used in all other experiments.

-Other factors that have been proposed to contribute to nucleoid organization, like macromolecular crowding (e.g., 31155353), solvent quality (e.g., PMID 34186018), electrostatic interactions (e.g., PMID 31629479), DNA supercoiling (e.g., PMID 3183003), or the activity of NAPs (e.g., 31767998) are

not considered nor discussed (while almost all of them might be altered during the chemical perturbations).

Response: We agree with the Reviewer that there are many cellular processes that regulate nucleoid organization. We thank the Reviewer for the selection of references, of which some were already cited in our manuscript and which – together with the possible mentioned mechanisms – we added to the discussion and contextualize with our findings.

Minor comments

-Vercini measurements: what are we looking at exactly, different cells? How was the imaging plane chosen? Z-stacks and time-lapse videos would be extremely interesting and relevant to add.

Response: We thank the Reviewer for highlighting this missing information. Shown are different cells that were acquired in HILO illumination. A z-stack was acquired with 0.5 μm step size. This large step-size was required to minimize photobleaching of mScarlet-I, which is known to be less photostable in bacteria compared to eukaryotes. As this provides only a few sections per nucleoid, and cells are trapped at different positions inside the VerCINI traps, we manually selected the z-slices of different cells. This is now indicated in the manuscript.

-Is Nile Red a proxy for the inner or outer membrane, or both?

Response: To our knowledge, Nile red integrates in both membranes. It was shown to flip across membranes and has no preference for different membranes in eukaryotic cells (<https://doi.org/10.1021/acs.nanolett.8b04385>).

-A quantification of to what extent MreB localization depends on the chosen deconvolution approach would be helpful to assess potential variability introduced by the chosen method.

Response: We tested our approach on raw MreB images and another deconvolution approach (Richardson-Lucy deconvolution) (Fig. S27). We observe the center peak in all three cases. Interestingly, the peak for raw data is also very prominent, which we attribute to the data normalization (z-score) which removes contribution of the blurry background signal.

-While many of the figures show averages and standard deviations, the underlying cell numbers are not indicated. Same for the number of replicate experiments.

Response: We added cell numbers to the text and Supplementary tables. Confocal imaging was performed in duplicate, showing high reproducibility. SMLM imaging was performed on one of these replicates.

-Line 274: does the mention of “relative DNA content” refer to the relative DNA amount or relative positioning of the DNA? Can the super-resolution microscopy provide insight into the absolute or relative amount of DNA inside cells?

Response: We thank the Reviewer for this excellent question. Unfortunately, we cannot make any statements on the DNA content based on signal intensity. One reason is that single-molecule imaging requires a good separation of binding events. For compacted nucleoids (e.g. chloramphenicol-treated cells or 5-10 min timepoints of rifampicin/chloramphenicol treatment) we had to reduce the concentration of PAINT labels in order to achieve this separation. As many other steps can slightly influence quantitative nature (e.g. permeabilization strength, effective concentration of PAINT labels), we aimed for a structural study. We think that the method can in principle be converted into a quantitative approach (very slow imaging; calibration samples, etc.), yet there are easier

approaches to determine DNA content for varying conditions (e.g. FACS analysis).

-Line 441-445: It was previously shown that cell size correlates with nucleoid size, but inversely with nucleocytoplasmic ratio across species, leading to more nucleoid exclusion of ribosomes (PMID 31150626). This appears to directly oppose the hypothesis of the authors.

Response: We thank the Reviewer for pointing out this important study that we also cite in our manuscript. Our results do not contradict this finding. As we show in Figure S4, the nucleoid distribution appears to be constant across different length intervals. This means that nucleoid length (which can be used as a proxy for nucleoid size here) correlates with cell size, as stated by the Jacobs-Wagner lab. We cannot make a statement on the correlation across species, but this would be an exciting follow-up project.

-Movement of gene loci to the nucleoid periphery has been shown to depend on transcription and translation, but not on transertion (PMID 31719538). Would this not argue against a strong role for transertion in nucleoid organization?

Response: We thank the Reviewer for highlighting the study. We added this reference to the manuscript and included their results in our discussion. The results that transcription and translation alone mediate repositioning are very interesting, but do not rule out the presence or role of transertion. It can be argued that the positioning of genetic loci at the nucleoid surface could even be a requirement for transertion, as the genes/mRNAs need to be positioned close to the membrane in order to install the dynamic links. We also note that the growth conditions in the mentioned study differ from ours. Mass-doubling times in this study were 55 or 99 min, while our cultures showed a mass doubling time of 27 min. As shown in Figure 1, the growth rate strongly effects nucleoid structure and positioning. This makes the comparison with our results difficult.

Response to the reviewers

Reviewer #1 (Remarks to the Author):

We appreciate that the authors have performed additional experiments with RNaseE overexpression and provided more rigorous quantification of the DNA-membrane distances and of the MreB-DNA distances. The results match their conclusions better now. We find their responses to our concerns generally thorough and satisfactory. One major suggestion is to use “membrane proximity” or “close to membrane” instead of “membrane-associated” in the manuscript to avoid confusions. For example, the title could be revised as: The nucleoid of rapidly growing *Escherichia coli* localizes close to the inner membrane and is organized by transcription, translation and cell geometry

Response: We thank the reviewer for the positive assessment of our revision. We replaced terms like “membrane-associated” by more indirect statements and also adjusted the title accordingly.

Minor Comments:

1. Line 177: Thank you for providing a citation explaining the relationship between sodium azide and protein translocation. However, nalidixate should not be described as primarily disrupting DNA replication. It more broadly changes the supercoiling distribution which affects overall chromosome structure and transcription, in addition to replication. Another note is that Nalidixates and Novobiocin both work for Gram negative cells, but with different mechanisms.

Response: We thank the reviewer for this clarification. We now changed it accordingly in the manuscript, referring to the inhibition of supercoiling. We also acknowledge the note about Novobiocin, which we will use in further studies.

2. Line 418: Additional discussion of the RNase E overexpression results would be helpful. Is it possible that actively translating mRNAs are protected from degradation despite RNase E overexpression?

Response: We added further discussion regarding the RNase E experiment (lines 483 – 486). It can well be that active translation protects mRNAs from degradation. However, this requires in-depth studies of mRNA stability over a range of RNase E expression levels. We thank the reviewer for this suggestion, it is an interesting route that we plan to investigate in more detail in the future.

Reviewer #1 (Remarks on code availability):

The ReadMe provided enough detail to use the code. One additional improvement would be to include an example file of data, but this is not necessary for other people to use the code.

Response: We thank the reviewer for the positive feedback. We provide the images and regions of interest used in this study via our Zenodo repositories. Thus, example data is already available.

Reviewer #2 (Remarks to the Author):

The authors have addressed all of our main comments and have made valuable additions to the manuscript. In particular, the authors now provide a more thorough discussion of the biophysical properties demonstrated by their experiments, and how they relate to the role of transertion. The greater discussion of the diverging results in M9 and LB is appreciated, as is the additional discussion of rifampicin and chloramphenicol phenotypes at longer treatment times. The amended title now better describes the claims supported by the work.

Some additional comments: the authors should clarify their reasoning in L542-546, in discussing the results of the RNase E expression experiments. Currently, the paragraph claims that transertion-mediated positioning is possible, and that RNase E localization and abundance does not significantly contribute to nucleoid organization, but the biological link between these statements should be made clearer in this discussion. The responses to comments 2 and 10 also need to be introduced in the ms.

Response: We thank the reviewer for the positive feedback on our revised manuscript. We now added further discussion of the RNase E experiment. In particular, we refer to the increased activity of RNase E on operons and mRNAs encoding membrane and periplasmic proteins. Increased degradation of such transcripts would weaken transertion, potentially leading to changes in nucleoid organization. However, it is not possible to quantify the effect on transertion and to estimate the extent of transertion inhibition that would lead to observable changes in nucleoid organization. Responses to comment 10 was already discussed in the revised manuscript (lines 465 - 470). We added additional explanations for comment 2 (lines 281 – 283).

Reviewer #3 (Remarks to the Author):

Response: We thank the reviewer for co-reviewing our manuscript, providing valuable feedback that helped us to improve this manuscript.

Reviewer #4 (Remarks to the Author):

I commend the author's extensive revisions and clarifications. However, I think that several of my main points still stand:

There is no direct evidence that implicates transertion. While the RNaseE experiments are interesting, I was hoping to see a larger effect. For now, these do not add much, besides, as the authors indicate in the study, that “degradosome activity, at least under the condition tested, does not significant contribute to nucleoid organization” (so not much transertion-related insights). Early timepoints of drug treatments can also not be seen as targeted disruptions of transertion. For now, transertion thus remains a hypothesis, and statements regarding it should be amended throughout the text. For example, subtitles such as “Super-resolution microscopy reveals complete nucleoid detachment from the inner membrane upon inhibition of protein biosynthesis” or statements in the discussion like “indicates that active transertion is required to maintain this expanded, membrane-localized state.” are somewhat misleading.

Similarly, other factors involved in nucleoid organization were not directly tested, so their suggested non-involvement also remains speculative.

Response: We further softened our statements and remove terms such as “membrane-associated” or “complete detachment” throughout the manuscript. As the reviewer mentioned, many factors involved in nucleoid organization, directly or indirectly. Testing them comprehensively is a major goal for our future research and will reveal further insights into nucleoid organization.

Related to specific contributions/novelty: although I agree with the authors that SMLM increases contrast and resolution, I still struggle to find new biological insights or knowledge that have come from using this approach, and that were previously not accessible from literature (were standard microscopy methods were mostly used).

Response: We thank the reviewer for the critical evaluation. We agree that literature provided similar results, particularly the study by the Weisshaar lab. However, the diffraction-limited images did not provide a detailed view on nucleoid structuring, nor on the extent of compaction/reorganization. Furthermore, we are not aware that the membrane-proximal positioning, together with the DNA-depleted radial cell center, were reported in literature before. Our super-resolution and VerCINI images clearly demonstrate this striking nucleoid phenotype that contradicts the classical view of chromatin distribution in *E. coli*. Next to the biological insights, our study provides new tools that can be used for the many other high-resolution studies in bacteria.

With regard to the apparent association of MreB with chromosomal DNA, many of my points were not addressed in the rebuttal:

-In Figure 4G, the circular cross-correlation for shorter cells displays a smaller peak than that observed in larger cells, arguing in favor of my previous argument that the apparent co-localization simply stems from both components localizing to the same cellular region (i.e., not the poles nor the division septum).

Response: The differences in peak heights could also be interpreted as a more prominent co-localization caused by existence of two sister nucleoids in long cells. Measuring the circular cross-correlation only along nucleoid areas would not provide meaningful results, as the intensity traces would be concatenated in an artifactual way. To us, a spatial correlation between the elongasome and the nucleoid still remains a likely cause of the observed peak. However, we agree that this correlation might be of indirect nature, e.g. due to transertion of components of the elongation complex. We also discuss this in our manuscript.

-Related, in Figure S25 (first column), there is a negative correlation between the cytosolic control and DNA, suggesting exclusion of a free cytosolic protein by the nucleoid. Is this correct, artefactual, or how do the authors interpret this? (given that exclusion of a small, free cytosolic protein by the nucleoid would be unexpected)

Response: The reviewer correctly spotted this phenomenon, which we also describe in the main text (“Cytosolic PAmCherry1 was excluded from the nucleoid region, leading to anti-correlated signal. This anti-correlation was picked up by our circular cross-correlation approach, yielding a negative peak centred at 0.”). A recent study (<https://doi.org/10.1073/pnas.2406340121>) showed that positive charges lead to exclusion of particles from the nucleoid. Although these particles

were larger in size, the positive charge (+6) on the His₆-PAmCh1 might explain the observed exclusion.

-I am still not sure how I should interpret the y-axis in terms of correlation strength. Is a value of 50 (circular x-corr [A.U.]) a lot? Why are these not expressed as interpretable correlation coefficients?

Response: Our approach uses z-normalized intensity traces to enable cross-correlation on data with varying signal strengths, and reports on the spatial context of the signals' correlation. This results in a correlation with arbitrary units instead of classical correlation coefficient. We thus cannot quantify colocalization, but we can qualitatively state that the cross-correlation between MreB and the nucleoid is smaller than between RNA polymerase and the nucleoid (x-corr values of 50 – 100 compared to 400 – 500). This is well expected, as >80% of RNAP are associated with the nucleoid under the conditions used, decorating most of the nucleoid surface, while MreB only colocalizes to a fraction of the nucleoid. We now added this qualitative comparison to the manuscript (lines 342 – 348).

-Why would this association exist? Also, many more membrane protein assemblies exist, why did the author focus on this one?

Response: We agree that several other membrane protein assemblies exist. We chose to investigate MreB due to reported interactions with components of the transcription and translation machinery (see lines 487 - 489). Also, the elongasome is one of the largest, membrane-localized protein assemblies, eventually causing a biased localization of the chromosome to these assemblies.